# Translation of dipeptide repeat proteins in *C9ORF72* ALS/FTD through unique and redundant AUG initiation codons

Yoshifumi Sonobe[1,2,3], Soojin Lee[4,5], Gopinath Krishnan[4,5], Yuanzheng Gu[6], Deborah Y Kwon[6], Fen-Biao Gao[4,5], Raymond P Roos[1,2,3]*†, Paschalis Kratsios[3,7]*†

[1]University of Chicago Medical Center, Chicago, United States; [2]Department of Neurology, University of Chicago Medical Center, Chicago, United States; [3]Neuroscience Institute, University of Chicago, Chicago, United States; [4]RNA Therapeutics Institute, University of Massachusetts Chan Medical School, Worcester, United States; [5]Department of Neurology, University of Massachusetts Chan Medical School, Worcester, United States; [6]Neuromuscular & Movement Disorders, Biogen, Cambridge, United States; [7]Department of Neurobiology, University of Chicago, Chicago, United States

**\*For correspondence:**
rroos@neurology.bsd.uchicago.edu (RPR);
pkratsios@uchicago.edu (PK)

†These authors contributed equally to this work

**Abstract** A hexanucleotide repeat expansion in *C9ORF72* is the most common genetic cause of amyotrophic lateral sclerosis (ALS) and frontotemporal dementia (FTD). A hallmark of ALS/FTD pathology is the presence of dipeptide repeat (DPR) proteins, produced from both sense GGGGCC (poly-GA, poly-GP, poly-GR) and antisense CCCCGG (poly-PR, poly-PG, poly-PA) transcripts. Translation of sense DPRs, such as poly-GA and poly-GR, depends on non-canonical (non-AUG) initiation codons. Here, we provide evidence for canonical AUG-dependent translation of two antisense DPRs, poly-PR and poly-PG. A single AUG is required for synthesis of poly-PR, one of the most toxic DPRs. Unexpectedly, we found redundancy between three AUG codons necessary for poly-PG translation. Further, the eukaryotic translation initiation factor 2D (EIF2D), which was previously implicated in sense DPR synthesis, is not required for AUG-dependent poly-PR or poly-PG translation, suggesting that distinct translation initiation factors control DPR synthesis from sense and antisense transcripts. Our findings on DPR synthesis from the *C9ORF72* locus may be broadly applicable to many other nucleotide repeat expansion disorders.

## Editor's evaluation

This study by Sonobe et al. uses transfected cells and patient iPSC-derived neurons to define mechanisms underlying translation of the antisense CCCCGG RNA strand expressed in *C9ORF72*-associated ALS and FTD. The authors design a series of constructs to explore the start codon required to produce toxic PR and prominent PG dipeptides in disease. Using these constructs they provide solid data that translation in the PR and PG reading frames occurs due to the presence of AUG codons within the 5'UTR of the RNA strand.

## Introduction

The hexanucleotide GGGGCC repeat expansion in the first intron of *C9ORF72* is the most common monogenic cause of inherited amyotrophic lateral sclerosis (ALS) and frontotemporal dementia (FTD) (*DeJesus-Hernandez et al., 2011*; *Renton et al., 2011*). This mutation is predicted to cause ALS/FTD via three non-mutually exclusive mechanisms: (1) a loss-of-function mechanism due to reduced

C9ORF72 protein expression (*Liu et al., 2022b*; *Banerjee et al., 2023*; *Dane et al., 2023*; *Zhu et al., 2020*), (2) a gain-of-function mechanism due to toxicity from repeat-containing sense (GGGGCC) and antisense (CCCCGG) RNA (*McEachin et al., 2020*; *Parameswaran et al., 2022*), and (3) toxicity from dipeptide repeat (DPR) proteins produced from these transcripts (*Loveland et al., 2022*; *Taylor et al., 2016*; *Kwon et al., 2014*; *Wen et al., 2014*). However, loss of C9ORF72 protein by itself does not cause neurodegeneration (*Koppers et al., 2015*). On the other hand, DPRs produced from both sense (poly-GA, poly-GP, poly-GR) and antisense (poly-PR, poly-PG, poly-PA) transcripts are present in the central nervous system of ALS/FTD patients (*Zu et al., 2013*; *Gendron et al., 2013*). Strong evidence from experimental model systems suggests DPRs are toxic (*Schmitz et al., 2021*), underscoring the importance of uncovering the molecular mechanisms responsible for DPR synthesis.

To design therapies that reduce DPR levels, it is valuable to identify initiation codons used in DPR translation. To date, the synthesis of sense DPRs has been a major focus in the ALS/FTD field, resulting in the identification of translation initiation codons for poly-GA and poly-GR (*Green et al., 2017*; *Tabet et al., 2018*; *Boivin et al., 2020*; *Sonobe et al., 2018*). As previously shown, *non-canonical* codons (CUG for poly-GA, AGG for poly-GR) initiate DPR synthesis from the sense strand (*Green et al., 2017*; *Tabet et al., 2018*; *Boivin et al., 2020*; *Sonobe et al., 2018*; *van 't Spijker et al., 2022*). Interestingly, studies in *Drosophila* and cultured cells showed that the presence of an expanded GGGGCC repeat alone, without flanking intronic sequences, can result in DPR production, suggesting an unconventional form of translation (*Zu et al., 2013*). However, deletion analysis of *cis*-regulatory elements upstream of the GGGGCC repeats and ribosome profiling revealed that translation initiation in the poly-GA and poly-GR frames does depend on flanking intronic sequences surrounding the repeats (*van 't Spijker et al., 2022*; *Lampasona et al., 2021*; *Almeida et al., 2019*). Moreover, a recent study proposed that a canonical AUG initiation codon is used for poly-PG synthesis from the antisense CCCCGG transcript (*Boivin et al., 2020*), suggesting conventional translation is involved in the synthesis of at least one DPR. However, the initiation codons for other DPRs (e.g., poly-PR, poly-PA) from the antisense transcript remain unknown. Hence, it is unclear which mode of translation is utilized for DPR synthesis from the antisense transcript.

Although both sense and antisense transcripts produce GP-containing dipeptides (sense: poly-GP, antisense: poly-PG), the antisense transcript seems to be the primary source of poly-PG/poly-GP inclusions in the brain of *C9ORF72* ALS/FTD patients (*Zu et al., 2013*). Further, two recent ALS clinical trials that specifically targeted the production of DPRs from the sense transcript failed (*Liu et al., 2022a*; *Tran et al., 2022*; *Krishnan et al., 2022*). Therefore, studying the mechanisms responsible for DPR synthesis from the antisense transcript is important, and this is the focus of the present study.

An additional challenge in ALS/FTD is the identification of regulatory factors necessary for DPR synthesis. Research efforts have uncovered a number of proteins that act at different steps of DPR synthesis: RNA helicases (eIF4A, DHX36, and DDX3X) (*Green et al., 2017*; *Tseng et al., 2021*; *Cheng et al., 2019*), proteins of the eIF4F complex (eIF4A, eIF4B, eIF4E, eIF4H) (*Green et al., 2017*; *Cheng et al., 2018*; *Goodman et al., 2019*; *Linsalata et al., 2019*), small ribosomal protein subunit 25 (RPS25) (*Yamada et al., 2019*), ribosome quality control protein ZNF598 (*Park et al., 2021*), and eukaryotic translation initiation factors (DAP5 [*van 't Spijker et al., 2022*], eIF2A [*Sonobe et al., 2018*], eIF3F [*Ayhan et al., 2018*], eIF2D [*Sonobe et al., 2021*], and eIF2D co-factors DENR and MCTS-1 [*Green et al., 2022*]). Except RPS25, all remaining factors have only been assessed for their effects on DPRs produced from the sense GGGGCC transcript. Furthermore, the role of these factors on DPR synthesis in induced pluripotent stem cell (iPSC)-derived neurons from *C9ORF72* ALS/FTD patients remains largely untested.

Here, we employ cell-based models of *C9ORF72* ALS/FTD to identify translation initiation codons for DPRs produced from the antisense transcript. Transfection into cultured cells of constructs carrying 35 CCCCGG repeats (preceded by 1000 bp of human intronic *C9ORF72* sequence) leads to DPR production (poly-PR, poly-PG) and reduced cell survival. We find that a canonical AUG initiation codon located 273 base pairs (–273 bp) upstream of the CCCCGG repeats is necessary for poly-PR synthesis. Further, we provide evidence for redundancy in usage of canonical initiation codons for poly-PG synthesis. Although an AUG at –194 bp is the main start codon for poly-PG, two other AUG codons (at –212 bp and at –113 bp) can also function as alternative translation initiation sites. These findings suggest that DPR synthesis from the antisense transcript occurs via AUG-dependent translation, contrasting with the mode of DPR synthesis from the sense transcript, which depends on

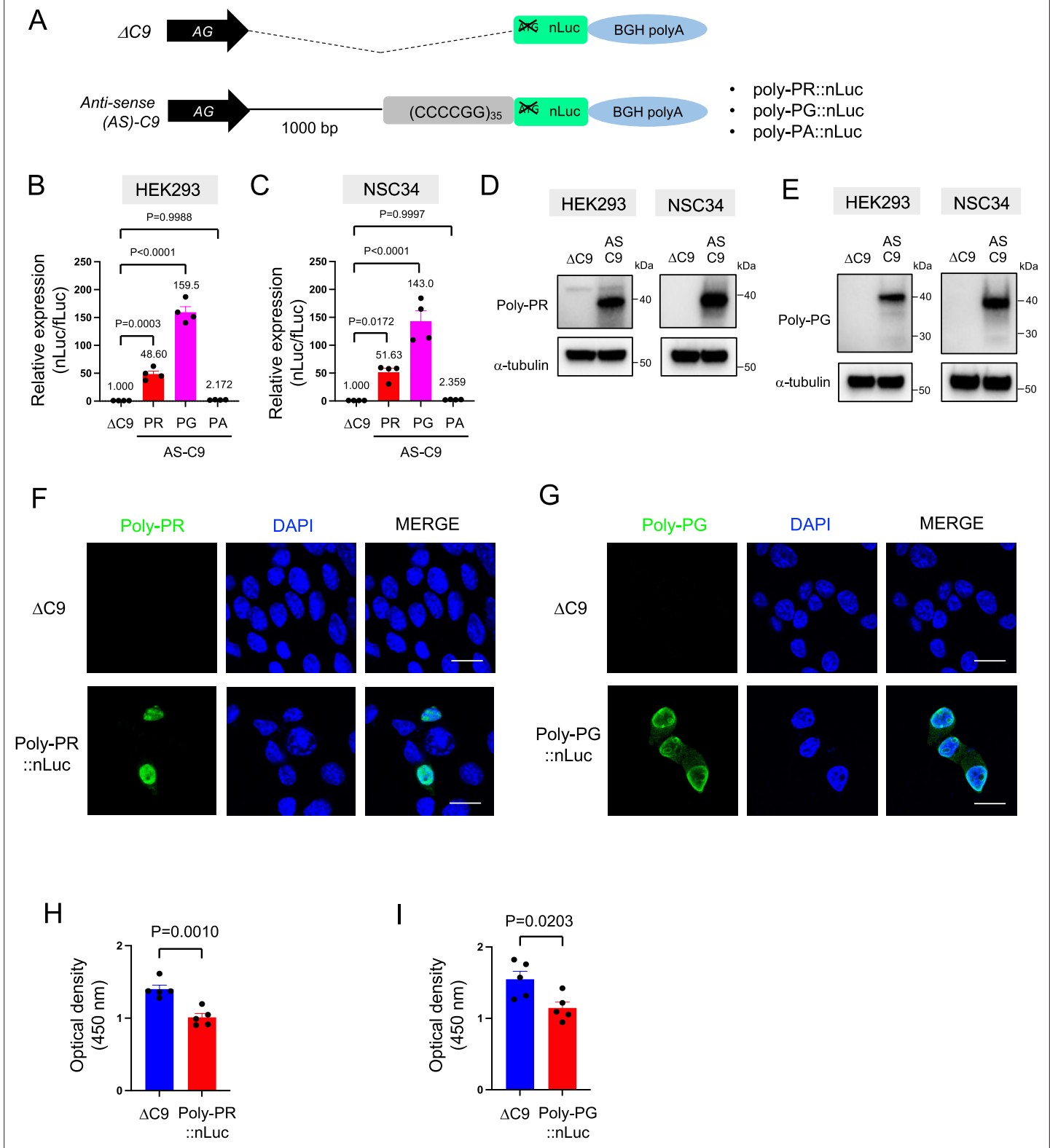

**Figure 1.** Poly-PR and poly-PG are translated from antisense CCCCGG repeats. (**A**) Schematic diagram of the constructs with 35 CCCCGG repeats preceded by 1000-bp-long intronic sequence from human *C9ORF72*, and then followed by nanoluciferase (nLuc). (**B**) HEK293 and (**C**) NSC34 cells were cotransfected with fLuc along with either ΔC9 or AS-C9 plasmids. The levels of luciferase activity were assessed by dual luciferase assays (mean ± s.e.m.). The experiments were repeated four times. One-way ANOVA with Tukey's multiple comparison test was performed. (**D–E**) HEK293 and NSC34 cells were transfected with either ΔC9 or AS-C9 plasmids. Cell lysates were processed for western blotting, and immunostained with antibodies to (**D**) poly-PR,

*Figure 1 continued on next page*

*Figure 1 continued*

(**E**) poly-PG, and α-tubulin. (**F–G**) NSC34 cells transfected with either ΔC9, (**F**) poly-PR::nLuc, or (**G**) poly-PG::nLuc were stained with a nuclear marker (4′,6-diamidino-2-phenylindole [DAPI]: blue) and with antibodies against poly-PR (F: green) or poly-PG (G: green). Scale bars indicate 20 µm. (**H–I**) NSC34 cells were transfected with either ΔC9, (**H**) poly-PR::nLuc, or (**I**) poly-PG::nLuc plasmids. WST-8 assay was performed to assess the cell viability. The experiments were repeated five times. Unpaired t test was performed.

The online version of this article includes the following source data and figure supplement(s) for figure 1:

**Source data 1.** Full raw unedited images of western blots shown in *Figure 1*.

**Figure supplement 1.** Nanoluciferase (nLuc) is fused to dipeptide repeats (DPRs) translated from antisense C9 plasmids containing CCCCGG repeats.

**Figure supplement 1—source data 1.** Full raw unedited images of western blots shown in *Figure 1—figure supplement 1*.

**Figure supplement 2.** Expression levels of poly-PR and poly-PG in the RIPA-insoluble fraction.

**Figure supplement 2—source data 1.** Full raw unedited images of western blots shown in *Figure 1—figure supplement 2*.

**Figure supplement 3.** Poly-PA is not detected by western blotting upon transfection of antisense C9 plasmids containing CCCCGG repeats.

**Figure supplement 3—source data 1.** Full raw unedited images of western blots shown in *Figure 1—figure supplement 3*.

non-canonical start codons (CUG for poly-GA, AGG for poly-GR). Finally, we show that the translation initiation factor eIF2D, which is necessary for CUG-dependent poly-GA synthesis from the sense transcript (*Sonobe et al., 2021*), is not involved in AUG-dependent antisense DPR (poly-PG, poly-PR) synthesis. Hence, distinct translation initiation sites and factors are employed for DPR synthesis from sense GGGGCC and antisense CCCCGG transcripts.

## Results

### Transfection of constructs carrying 35 CCCCGG repeats leads to antisense DPR synthesis and reduced cell survival

To study DPR synthesis from the antisense transcript, we engineered three constructs with 35 CCCCGG repeats preceded by 1000-bp-long intronic sequence from human *C9ORF72* (*Figure 1A*; *Sonobe et al., 2021*), and then followed by nanoluciferase (nLuc) in frame of poly-PR, poly-PG, or poly-PA (see Materials and methods). 48 hr after transfection of poly-PR::nLuc or poly-PG::nLuc into HEK293 and NSC34 cells, robust expression of poly-PR and poly-PG was detected both in luciferase assays (*Figure 1B–C*) and western blotting for poly-PR, poly-PG, and nLuc (*Figure 1D–E*, *Figure 1—figure supplement 1*, *Figure 1—source data 1*), suggesting the luciferase signal is an accurate readout for DPR production. Protein isolation of soluble and insoluble fractions showed that both DPRs (poly-PG and poly-PR) are predominantly detected in the soluble fraction under these experimental conditions (*Figure 1—figure supplement 2*). Further, production of poly-PR and poly-PG in transfected NSC34 cells was confirmed with immunofluorescence staining (*Figure 1F–G*). Finally, transfection of either poly-PR::nLuc or poly-PG::nLuc into NSC34 cells led to reduced cell survival (*Figure 1H–I*).

Consistent with a previous study (*Boivin et al., 2020*), we did not detect poly-PA with luciferase assays (*Figure 1B–C*) and western blotting (*Figure 1—figure supplement 3*) upon poly-PA::nLuc transfection. We surmise that the initiation codon for poly-PA may lie outside the 1000 bp intronic sequence used in our construct, or that the specific regulatory machinery needed for poly-PA synthesis is lacking in the cellular context examined here (HEK293 and NSC34 cells). Altogether, our cell-based model of *C9ORF72* (construct with 35 CCCCGG repeats and 1000 bp of human intron) produces two antisense DPRs (poly-PR, poly-PG) and displays reduced cell survival.

### A canonical AUG initiation codon located 273 bp upstream of CCCCGG repeats is required for poly-PR synthesis

The poly-PR::nLuc and poly-PG::nLuc constructs offer an opportunity to identify the initiation codons for poly-PR and poly-PG synthesis. We initially focused on poly-PR, one of the most toxic DPRs based on in vitro (*Kwon et al., 2014*; *Lee et al., 2016*; *Lin et al., 2016*) and in vivo studies in worms (*Rudich et al., 2017*), flies (*Wen et al., 2014*; *Lee et al., 2016*; *Maor-Nof et al., 2021*), and mice (*Maor-Nof et al., 2021*; *Zhang et al., 2019*; *Hao et al., 2019*). Using our recently developed machine-learning algorithm for initiation codon prediction (*Gleason et al., 2022*), we identified a CUG at –366 bp (Kozak sequence: guaCUGa) and an AUG at –273 bp (Kozak sequence: cggAUGc) as putative

initiation codons for poly-PR (*Figure 2A*). We then mutated these codons either to CCC or the termination codon UAG (*Figure 2A*). Western blotting and luciferase assays showed that mutation of the CUG at −366 bp to CCC or UAG did not affect poly-PR expression (*Figure 2B–G*, *Figure 2—source data 1*). However, mutation of the AUG at −273 bp to CCC or UAG completely abolished poly-PR expression both in HEK293 and NSC34 cells, as shown by western blotting (*Figure 2B–E*), luciferase assays (*Figure 2F–G*), and immunofluorescence staining against poly-PR (*Figure 2H*). Importantly, the reduced survival of NSC34 cells upon poly-PR::nLuc transfection was partially rescued when the −273 bp AUG codon was mutated into the UAG termination codon, suggesting poly-PR production is toxic under these experimental conditions (*Figure 2I*). These results strongly suggest that the AUG at −273 bp is the start codon for translation of poly-PR, one of the most toxic DPRs in *C9ORF72* ALS/FTD. This AUG is predicted to be included in the endogenous antisense CCCCGG transcript based on 5' Rapid Amplification of cDNA Ends (RACE) analysis on brain samples of *C9ORF72* ALS/FTD patients (*Zu et al., 2013*).

## Evidence for redundancy of AUG initiation codon usage in poly-PG translation

We next investigated poly-PG, which is less toxic than poly-PR (*Wen et al., 2014*; *Lee et al., 2016*; *Mizielinska et al., 2014*; *Freibaum et al., 2015*), and has been proposed as a biomarker for *C9ORF72*-ALS/FTD (*Gendron et al., 2017*; *Lehmer et al., 2017*). Using the same machine-learning algorithm (*Gleason et al., 2022*), we identified four putative initiation codons (AUG at −212 bp, AUG at −194 bp, CUG at −182 bp, AUG at −113 bp) (*Figure 3A*), all with relatively good Kozak sequences (gaaAUGa at −212 bp, aaaAUGc at −194 bp, gctCUGa at −182 bp, aggAUGc at −113 bp). Of note, a prior publication previously identified the AUG at −194 bp as an initiation codon (*Boivin et al., 2020*). Simultaneous mutation of all four of these codons to CCC completely blocked poly-PG expression (*Figure 3B–D*, *Figure 3—source data 1*), suggesting one or more of these codons is required. Next, we simultaneously mutated three codons to CCC, but left intact the AUG at −212 bp. We refer to this construct as '−212 AUG'. Upon transfection of −212 AUG, we observed poly-PG expression, suggesting poly-PG translation can start at the AUG at −212 bp. Intriguingly, when we followed a similar approach to mutate three codons to CCC but leave intact the AUG at −194 bp or at −113 bp, we also observed poly-PG production, but this time at an expected lower molecular weight (*Figure 3B–D*, *Figure 3—source data 1*). Of note, when we mutated to CCC all three AUG codons (−212 bp, −194 bp, −113 bp) but left intact the CUG at −182 bp, we observed no poly-PG expression (*Figure 3B–D*, *Figure 3—source data 1*). These results suggest that any of these three AUGs, but not the CUG at −182 bp, can function as a start codon for poly-PG, indicating redundancy in the translation initiation codon for poly-PG.

We observed a strong (higher molecular weight) band and a fainter (lower molecular weight) band for poly-PG when the intact version of the poly-PG::NanoLuc plasmid was translated (*Figure 3B*, *Figure 3—figure supplement 1*, *Figure 3—source data 1*). The strong band is likely to result from translation initiation at the AUG at −194 bp, whereas the faint band is likely initiated at the AUG at −113 bp (*Figure 3B*). Hence, the AUG at −194 bp appears to be the main initiation codon for poly-PG synthesis from the antisense transcript of 35 CCCCGG repeats (*Figure 3B*), which is consistent with mass spectrometry results from a previous report (*Boivin et al., 2020*).

Interestingly, selective mutation of the AUG at −194 to CCC did not abolish poly-PG expression (*Figure 4A–D*, *Figure 4—figure supplement 1*). Instead, it led to the production of two poly-PG products: a high molecular weight product (strong band) resulting from use of the AUG at −212 bp as well as a lower molecular weight product (faint band) resulting from AUG at −113 bp (*Figure 4B*, *Figure 4—source data 1*). Altogether, these results suggest that the AUG at −194 bp is mainly used for poly-PG expression from antisense CCCCGG repeats. However, when this AUG is mutated, two other AUG codons (at −212 bp and −113 bp) can also function as translation initiation sites, again revealing redundancy in the start codon usage for poly-PG synthesis.

## Mutation of the −113bp AUG abolishes poly-PG production

We further corroborated this redundant initiation of poly-PG translation by individually mutating each of the AUG codons to a termination UAG codon (*Figure 5A–D*, *Figure 5—figure supplement 1*, *Figure 5—source data 1*). Mutation of the AUG at −212 bp to UAG (construct name: −212 UAG) did

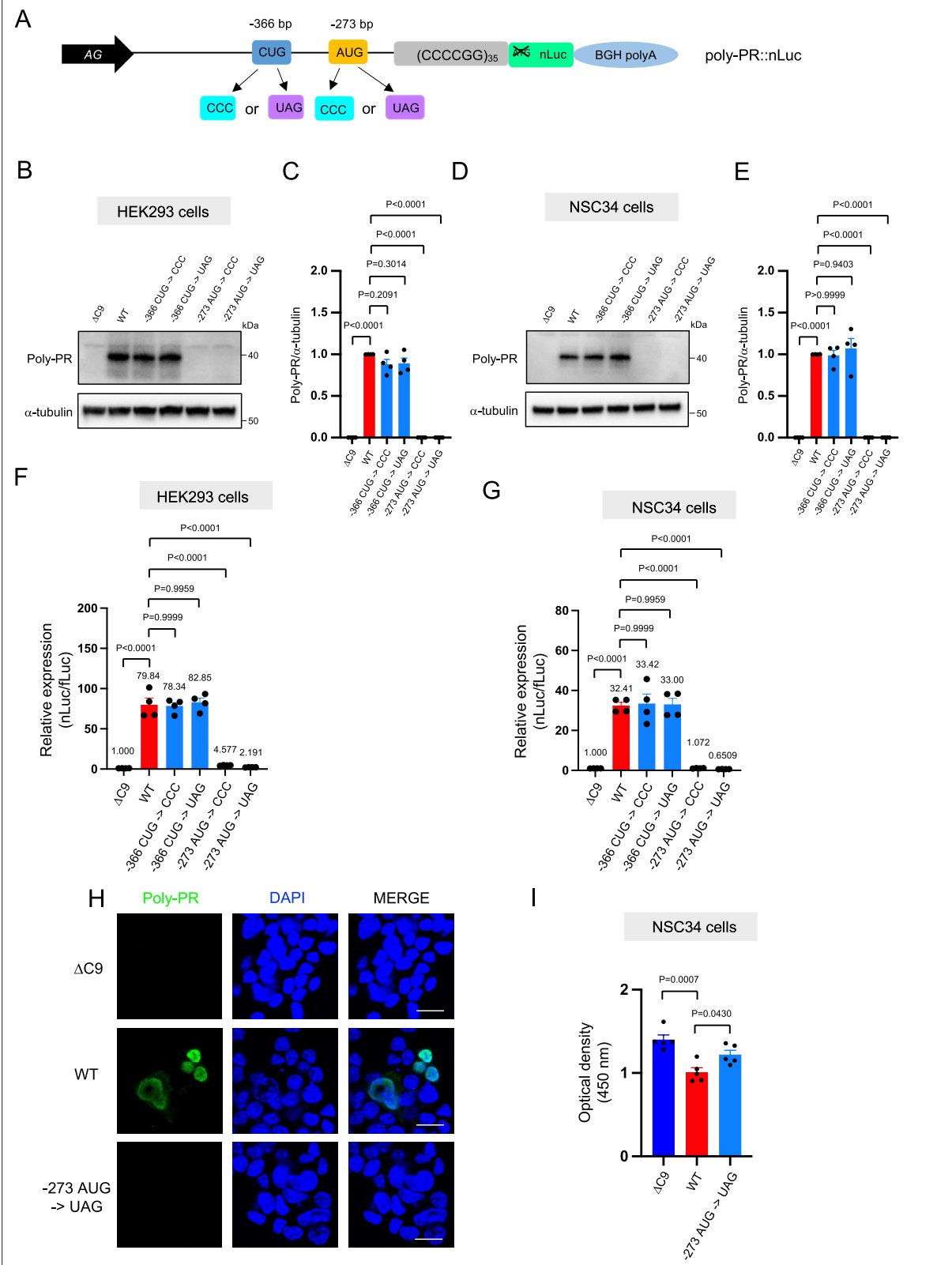

**Figure 2.** An AUG at –273 bp position is the start codon for poly-PR translation. (**A**) Schematic diagram showing constructs with mutations in the putative start codons for poly-PR. HEK293 (**B–C**) and NSC34 (**D–E**) cells were transfected with indicated plasmids. Cell lysates were processed for western blotting, and immunostained with antibodies to poly-PR and α-tubulin. (**B, D**) Representative blots are shown. (**C, E**) The signal intensity of the bands were quantified (mean ± s.e.m.). The experiments were repeated four times. One-way ANOVA with Tukey's multiple comparison test was

*Figure 2 continued on next page*

*Figure 2 continued*

performed. (**F**) HEK293 and (**G**) NSC34 cells were cotransfected with the plasmids along with fLuc. The levels of luciferase activity were assessed by dual luciferase assays (mean ± s.e.m.). The experiments were repeated four times. One-way ANOVA with Tukey's multiple comparison test was performed. (**H**) NSC34 cells transfected with either ΔC9, poly-PR::nLuc, or –273 AUG ->UAG plasmids were stained with 4',6-diamidino-2-phenylindole [DAPI] (blue) and immunostained with a poly-PR antibody (green). Scale bars show 20 μm. (**I**) NSC34 cells were transfected with either ΔC9, wild type (WT), or –273 AUG ->UAG plasmids. WST-8 assay was performed to assess the cell viability. The experiments were repeated five times. One-way ANOVA with Tukey's multiple comparison test was performed. In ΔC9 and WT, the same datasets as *Figure 1H* were used (mean ± s.e.m.). The experiments were repeated five times. One-way ANOVA with Tukey's multiple comparison test was performed.

The online version of this article includes the following source data for figure 2:

**Source data 1.** Full raw unedited images of western blots shown in *Figure 2*.

not affect poly-PG expression, most likely because the AUG at –194 bp became the start codon as shown by western blotting (*Figure 5B–D*, *Figure 5—source data 1*). Similarly, mutation of the AUG at –194 bp to UAG (construct name: –194 UAG) did not affect poly-PG expression because the AUG at –113 bp became the start codon (*Figure 5B–D*). However, mutation of AUG at –113 bp to UAG (construct name: –113 UAG) completely blocked poly-PG expression, as shown by western blotting (*Figure 5B*, *Figure 5—figure supplement 1*), luciferase assays (*Figure 5C–D*), and immunofluorescence staining (*Figure 5E*). Finally, the reduced survival of NSC34 cells was not rescued upon transfection of the –113 UAG construct, suggesting poly-PG production is not toxic under these experimental conditions (*Figure 5F*).

Altogether, these findings strongly suggest that the AUG at –194 bp is primarily used for poly-PG translation, but the other two AUG codons at –212 bp and –113 bp can also function as translation initiation sites under certain experimental conditions.

## EIF2D does not control poly-PR and poly-PG synthesis from the antisense transcript

Following the identification of AUG codons for translation initiation of poly-PR and poly-PG, we next sought to identify translation initiation factors necessary for synthesis of these antisense DPRs. We focused on EIF2D because we previously found it to be necessary for poly-GA synthesis from the sense GGGGCC transcript in *Caenorhabditis elegans* and cell-based models (HEK293 and NSC34 cell lines) (*Sonobe et al., 2021*). To this end, we generated an *EIF2D* knockout HEK293 line using CRISPR/Cas9 gene editing (see Materials and methods) (*Figure 6A–C*, *Figure 6—source data 1*). Next, we transfected the poly-PR::nLuc reporter construct into control and *EIF2D* knockout HEK293 cells. We found that knockout of *EIF2D* did not affect the expression levels of the poly-PR::nLuc reporter (*Figure 6E*). We obtained similar results upon knockdown of *EIF2D* with a short hairpin RNA (shRNA) (*Figure 6H*), again suggesting that eIF2D is not required for poly-PR synthesis from antisense CCCGG transcripts. Lastly, knockout or knockdown (shRNA) of *EIF2D* in HEK293 cells transfected with poly-PG::nLuc did not decrease poly-PG expression based on a luciferase assay (*Figure 6D and G*). Hence, knockout or knockdown of *EIF2D* does not affect the production of two antisense DPR (poly-PR, poly-PG). On the other hand, knockdown of *EIF2D* did reduce the levels of poly-GA (*Figure 6I*), a DPR generated from sense RNA. The poly-GA reduction is consistent with our previous observations in a *C. elegans* model of *C9ORF72* ALS/FTD (*Sonobe et al., 2021*), albeit more modest - likely due to a technical reason (see legend of *Figure 6I*).

## Knockdown of EIF2D in human iPSC-derived motor neurons

We next tested whether EIF2D is required for DPR synthesis in a cellular context that maintains the endogenous human *C9ORF72* gene locus. We initially used one published iPSC line from a *C9ORF72* carrier (line 26#6), as well as an isogenic control line (26Z90) which had CRISPR/Cas9-mediated deletion of expanded GGGGCC repeats (*Lopez-Gonzalez et al., 2019*). The iPSC lines were differentiated into motor neurons as previously described (*Lopez-Gonzalez et al., 2016*). Repeated transfection of a small interfering RNA (siRNA) against *EIF2D* (*EIF2D*-siRNA-1), but not of a control scrambled siRNA, resulted in robust downregulation of *EIF2D* mRNA as assessed by RT-PCR (*Figure 7A*) and eIF2D protein analysis (*Figure 7—figure supplement 1*). The mRNA levels of eIF2A, a related initiation factor, remained unaltered, suggesting specificity in the siRNA effect. Despite this knockdown, an immunoassay (conducted in a blinded manner) failed to show any differences in the steady-state

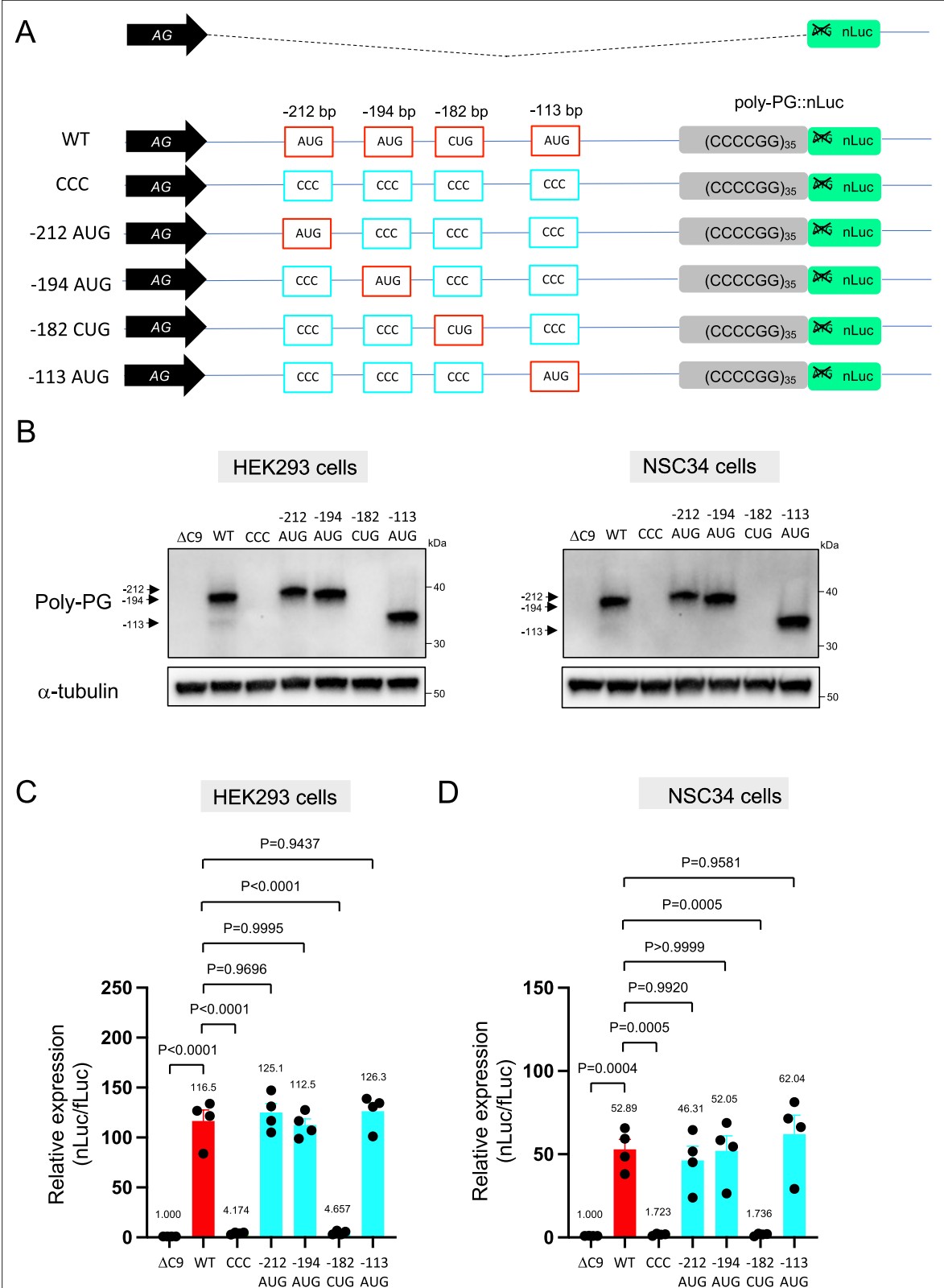

**Figure 3.** Mutation of AUG codons to CCC fails to suppress poly-PG translation. (**A**) Schematic diagram showing mutants with changes in the putative start codons for poly-PG. (**B**) HEK293 and NSC34 cells were transfected with indicated plasmids. Cell lysates were processed for western blotting, and immunostained with antibodies to poly-PG and α-tubulin. (**C**) HEK293 and (**D**) NSC34 cells were cotransfected with fLuc plasmid along with other

*Figure 3 continued on next page*

*Figure 3 continued*

indicated plasmids. The level of luciferase activity was assessed by dual luciferase assay (mean ± s.e.m.). The experiments were repeated four times. One-way ANOVA with Tukey's multiple comparison test was performed.

The online version of this article includes the following source data and figure supplement(s) for figure 3:

**Source data 1.** Full raw unedited images of western blots shown in *Figure 3*.

**Figure supplement 1.** Quantification of data from *Figure 3B*.

levels of soluble poly-PG (*Figure 7B*), suggesting eIF2D is not necessary for poly-PG translation from the antisense transcript. We caution though that our immunoassay does not distinguish between poly-PG produced from the antisense transcript and poly-GP from the sense transcript (*Figure 7B*). Hence, a mild effect upon EIF2D knockdown on poly-PG (from antisense transcript) can potentially be masked by poly-GP (from sense transcript). Of note, PG/GP inclusions in brain tissue of *C9ORF72* ALS/FTD patients contain ~80% of poly-PG from the antisense transcript and ~20% of poly-GP from the sense transcript (*Zu et al., 2013*). However, other studies indicate that the exact contribution of sense poly-GP and antisense poly-PG *C9ORF72* ALS/FTD has not been resolved (*Tran et al., 2022*;

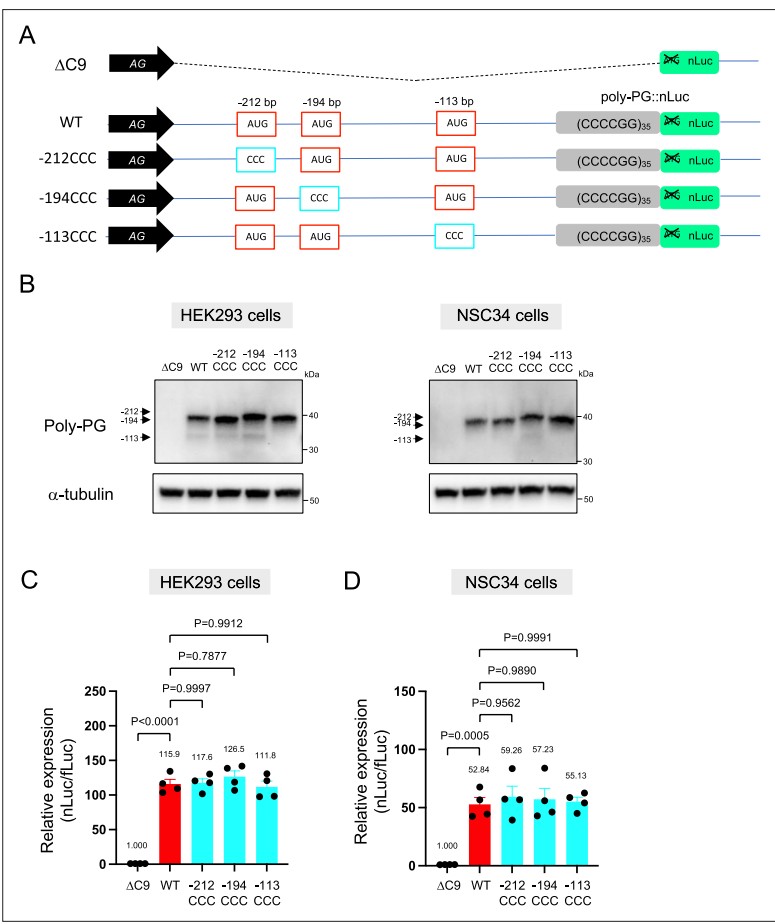

**Figure 4.** An AUG at −194 bp position is the primary start codon for poly-PG translation. (**A**) Schematic diagram of the constructs. (**B**) HEK293 and NSC34 cells were transfected with indicated plasmids. Cell lysates were processed for western blotting, and immunostained with antibodies to poly-PG and α-tubulin. (**C**) HEK293 and (**D**) NSC34 cells were cotransfected with fLuc plasmid along with indicated plasmids. The level of luciferase activity was assessed by dual luciferase assays (mean ± s.e.m.). The experiments were repeated four times. One-way ANOVA with Tukey's multiple comparison test was performed.

The online version of this article includes the following source data and figure supplement(s) for figure 4:

**Source data 1.** Full raw unedited images of western blots shown in *Figure 4*.

**Figure supplement 1.** Quantification of data from *Figure 4B*.

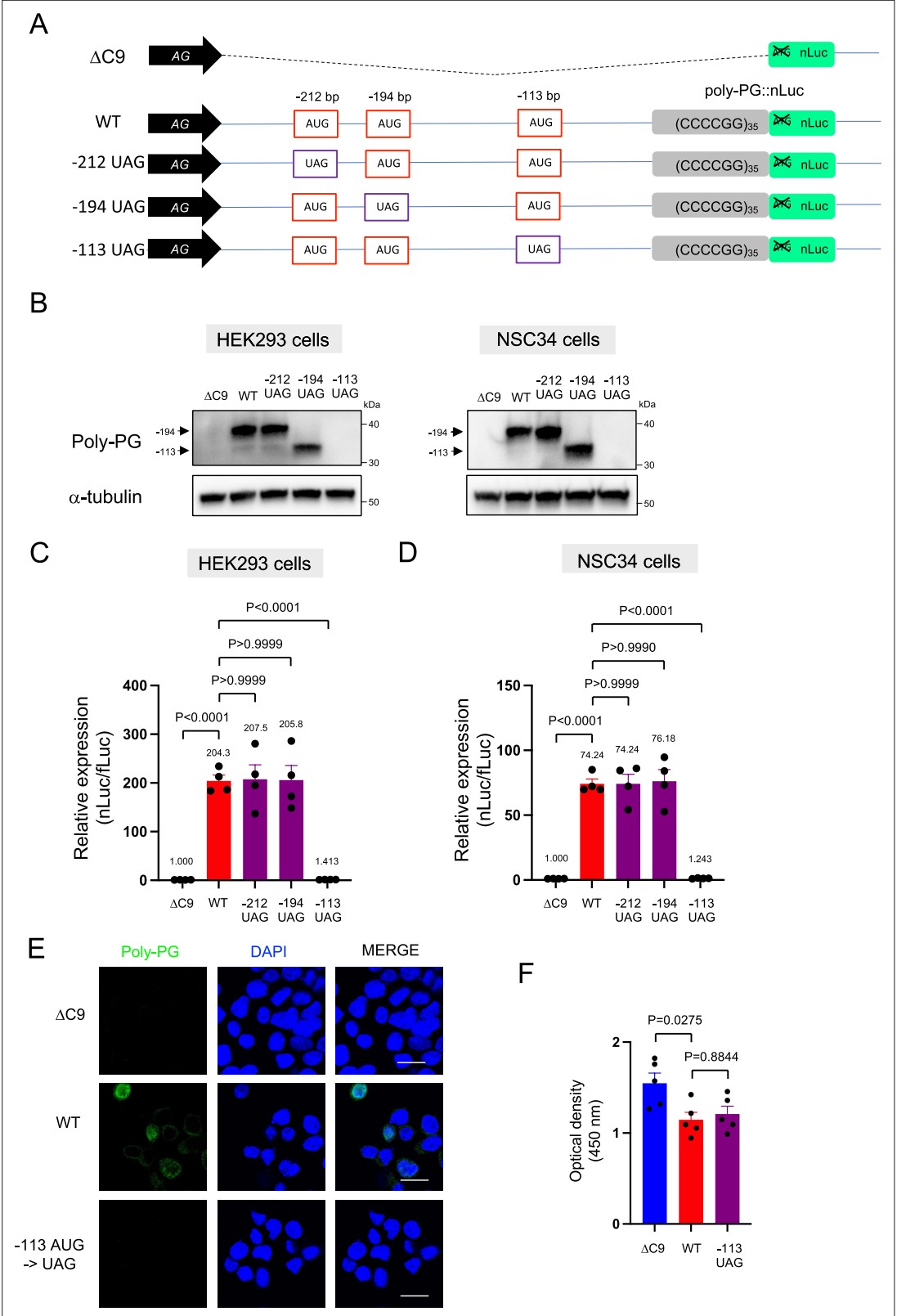

**Figure 5.** Redundancy of start codon usage in poly-PG translation. (**A**) Schematic diagram of the constructs. (**B**) HEK293 and NSC34 cells were transfected with indicated plasmids. Cell lysates were processed for western blotting, and immunostained with antibodies to poly-PG and α-tubulin. (**C**) HEK293 and (**D**) NSC34 cells were cotransfected with fLuc plasmid along with indicated plasmids. The level of luciferase activity was assessed by dual luciferase assays (mean ± s.e.m.). The experiments were repeated four times. One-way ANOVA with Tukey's multiple comparison test was performed.

*Figure 5 continued on next page*

*Figure 5 continued*

(**E**) NSC34 cells transfected with indicated plasmids were stained with 4',6-diamidino-2-phenylindole [DAPI] (blue) and immunostained with a poly-PG antibody (green). Scale bars show 20 μm. (**F**) NSC34 cells were transfected with indicated plasmids. WST-8 assay was performed to assess the cell viability (mean ± s.e.m.). The experiments were repeated five times. One-way ANOVA with Tukey's multiple comparison test was performed. In ΔC9 and wild type (WT), the same datasets as *Figure 1I* were used.

The online version of this article includes the following source data and figure supplement(s) for figure 5:

**Source data 1.** Full raw unedited images of western blots shown in *Figure 5*.

**Figure supplement 1.** Quantification of data from *Figure 5B and C*.

*Krishnan et al., 2022*; *Gendron et al., 2017*). Hence, our data hint that eIF2D may not affect poly-PG synthesis from the antisense CCCCGG transcript.

Despite the lack of an effect on poly-PG/GP, we found that *EIF2D* knockdown reduced poly-GA synthesis from the sense GGGGCC transcript in neurons derived from iPSC line 26#6 (*Figure 7B*), critically extending previous observations made in *C. elegans* and cell-based models (*Sonobe et al., 2021*). Consistent with the latter study, *EIF2D* knockdown had no effect on poly-GR synthesis from the sense transcript based on an immunoassay that measures soluble poly-GR (*Figure 7B*). Altogether, these findings from one patient line (26#6) suggest that eIF2D is required for CUG start codon-dependent poly-GA synthesis from the sense transcript in human iPSC-derived neurons, but is dispensable for poly-GR (from sense transcript) and poly-PG synthesis, albeit our immunoassay cannot distinguish between poly-PG and poly-GP. However, when we repeated this experiment with two additional iPSC lines (27#11 and 40#3) from *C9ORF72* carriers with two siRNAs (*EIF2D*-siRNA-1 and -2), we did not achieve robust *EIF2D* knockdown (*Figure 7C–D*). We note that the same siRNA (*EIF2D*- siRNA-1) led to robust *EIF2D* knockdown in the first patient line (26#6) (compare *Figure 7A* with *Figure 7C, D*). Hence, the issue of variable siRNA knockdown efficiency prevents us from drawing any general conclusions on the role of *EIF2D* in DPR synthesis in the context of motor neurons derived from different iPSC lines of *C9ORF72* carriers (*Figure 7B and E*).

## Discussion

Here, we show that canonical AUG codons on the antisense CCCCGG transcript serve as translation initiation codons for two DPRs - poly-PR and poly-PG. This finding may inform the design of future therapies for ALS/FTD, especially since poly-PR is a highly toxic DPR and poly-PG is thought to be primarily translated from the antisense transcript (*Zu et al., 2013*). Our finding of canonical AUG codons serving as translation initiation codons for antisense DPRs (poly-PR, poly-PG) differs from the proposed mode of translation of sense DPRs (e.g., poly-GA, poly-GR). In the latter case, it is thought that repeat-associated non-AUG (RAN) translation of poly-GA and poly-GR occurs via non-canonical CUG and AGG initiation codons, respectively, located in the intronic sequence upstream of the GGGGCC repeats (*Green et al., 2017*; *Tabet et al., 2018*; *Boivin et al., 2020*; *Sonobe et al., 2018*; *van 't Spijker et al., 2022*; *Sonobe et al., 2021*). Interestingly, studies in *Drosophila* and cultured cells showed that the presence of an expanded GGGGCC repeat alone, without flanking sequences, can result in DPR production (*Zu et al., 2013*; *Freibaum et al., 2015*). Hence, our findings together with these previous studies suggest that DPR synthesis may involve at least three different modes of translation: (1) near-cognate start codon (e.g., CUG, AGG) dependent translation for poly-GA and poly-GR from sense GGGGCC transcripts, (2) canonical AUG-dependent translation for poly-PR and poly-PG synthesis from antisense CCCCGG transcripts, and (3) DPR synthesis may also occur through RAN translation mechanisms that solely utilize the repeat. It is conceivable that all three modes of translation may occur simultaneously in disease, and that the use of non-canonical and canonical initiation codons may be the primary contributors of DPR production.

A notable finding is the presence of redundancy in start codon usage for poly-PG synthesis. Our data suggest that the AUG at –194 bp is primarily used for poly-PG translation from antisense CCCCGG transcripts, consistent with a previous investigation (*Boivin et al., 2020*). However, when this AUG is mutated, two other canonical AUG codons (at –212 bp and –113 bp) can also function as translation initiation sites under the experimental conditions described herein. Although it is

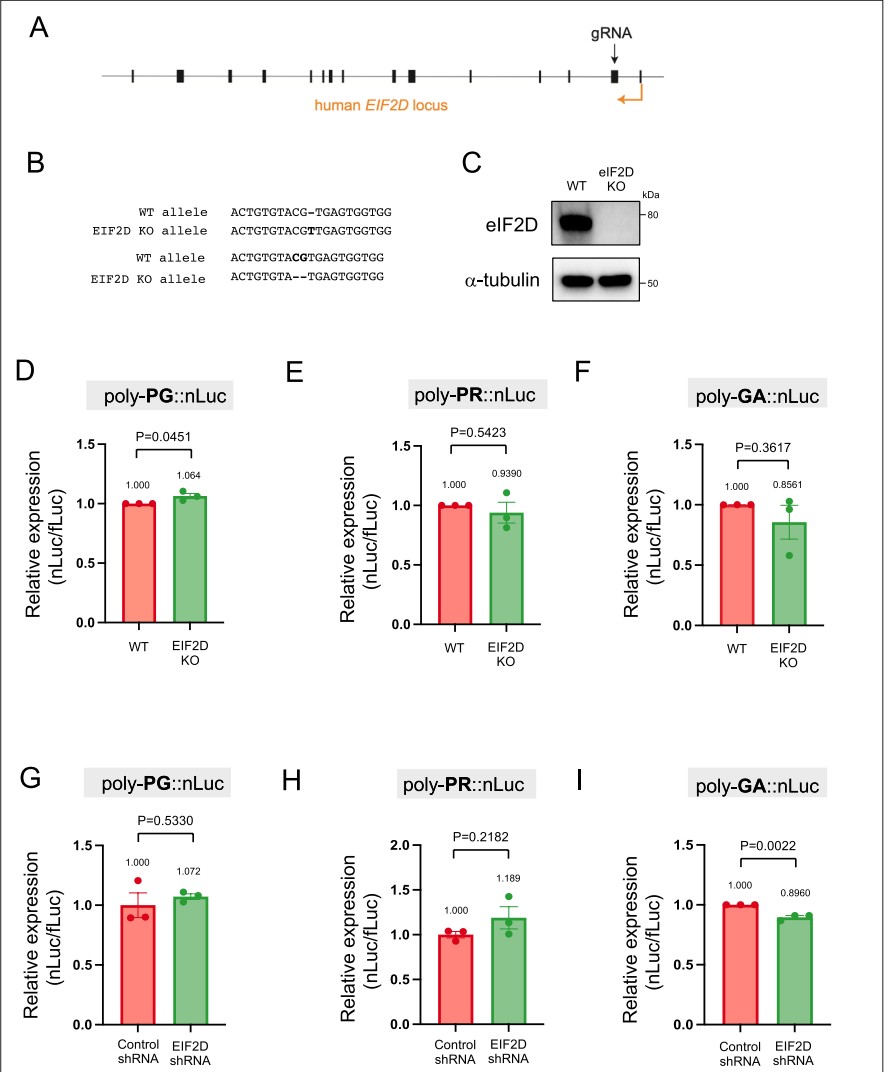

**Figure 6.** Downregulation of *EIF2D* does not reduce expression levels of poly-PG and poly-PR. (**A**) A gRNA targeted the second exon of human EIF2D (see Materials and methods). (**B**) After CRISPR/Cas9-mediated gene editing, the EIF2D knockout (EIF2DKO) HEK293 cells carried different mutations on each allele. (**C**) Cell lysates from wild type (WT) and EIF2DKO HEK293 cells were processed for western blotting, and immunostained with antibodies to eIF2D and α-tubulin. (**D–F**) WT and EIF2DKO HEK293 cells were cotransfected with fLuc plasmid along with either (**D–E**) AS-C9 plasmids or (**F**) C9 plasmids containing 75 GGGGCC repeats. The level of luciferase activity was assessed by dual luciferase assays. (**G–I**) WT HEK293 cells were transfected with fLuc and either (**G–H**) AS-C9 plasmids or (**I**) C9 monocistronic plasmids containing 75 GGGGCC repeats along with anti-EIF2D short hairpin RNA (shRNA). The level of luciferase activity was assessed by dual luciferase assays (mean ± s.e.m.). The experiments were repeated three times. Unpaired t test was performed. The poly-GA reduction upon EIF2D shRNA is consistent with our previous observations (*Sonobe et al., 2021*), albeit more modest - likely due to a technical reason (a bicistronic construct containing 75 GGGGCC repeats was used in *Sonobe et al., 2021*).

The online version of this article includes the following source data for figure 6:

**Source data 1.** Full raw unedited images of western blots shown in *Figure 6*.

unclear whether such redundancy in DPR translation initiation occurs in the central nervous system of *C9ORF72* ALS/FTD patients, these findings nevertheless suggest that targeting only one translation initiation site may be insufficient to prevent poly-PG synthesis. Redundancy in start codon usage may also apply to other DPRs, such as poly-PR synthesis from the antisense transcript. Although we identified an AUG at –273 bp as necessary for poly-PR synthesis, a previous study detected poly-PR when only 100 bp downstream of the GGGGCC repeats were included in an adeno-associated viral

(AAV) vector (*Chew et al., 2019*). It is important to note that this intronic 100-bp-long sequence was placed next to a 589 bp regulatory element of the woodchuck hepatitis virus (WPRE), which contains several putative start codons. The AUG initiation codons we identified as necessary for either poly-PR or poly-PG synthesis are predicted to be included in the endogenous antisense CCCCGG transcript based on 5' RACE analysis on brain samples of *C9ORF72* ALS/FTD patients (*Zu et al., 2013*). Nevertheless, endogenous mutagenesis of these codons - in the native genomic context of the *C9ORF72* locus - is needed in the future to further test the validity of our findings.

Emerging evidence suggests distinct proteins affect translation initiation of DPRs from sense and antisense transcripts in *C9ORF72* ALS/FTD. For example, the RNA helicase DDX3X directly binds to sense (GGGGCC), but not antisense (CCCCGG) transcripts, thereby selectively repressing the production of sense DPRs (poly-GA, poly-GP, poly-GR) (*Cheng et al., 2019*). Here, we provide evidence that the translation initiation factor EIF2D is not involved in DPR (viz., poly-PG, poly-PR) synthesis from antisense (CCCCGG) transcripts. In a previous study (*Sonobe et al., 2021*), we showed in *C. elegans* and in vitro cellular systems (HEK293 and NSC34 cells) that EIF2D is required for poly-GA production from sense (GGGGCC) transcripts. These findings are important because they indicate that not only distinct translation initiation codons, but also different regulatory proteins are involved in DPR synthesis from sense and antisense transcripts, suggesting that different modes of DPR translation (e.g., RAN translation, AUG-dependent translation) occur simultaneously in *C9ORF72* ASL/FTD. Consistent with this idea, translation initiation is the most heavily regulated step in protein synthesis because it is the rate-limiting step (*Richter and Sonenberg, 2005*). Hence, we favor a model where distinct regulatory factors are necessary for translation initiation of different DPRs. In striking contrast, the transcriptional control of sense and antisense transcripts appears coordinated. For example, a single protein - the transcription elongation factor Spt4 - controls production of both sense and antisense transcripts (*Kramer et al., 2016*).

In addition to *C9ORF72* ALS/FTD, nucleotide repeat expansions are present in various genes, causing more than 30 neurological diseases (*Chintalaphani et al., 2021*; *Depienne and Mandel, 2021*). In many of these, products translated from the expanded repeat sequences have been detected in the nervous system of affected individuals. Hence, our findings may also apply to this large group of genetic disorders in the following ways. First, translation of peptides from the same nucleotide repeat expansion may require different modes of translation (RAN- and AUG-dependent translation), as previously proposed (*Gao et al., 2017*). Second, the surprising redundancy in canonical AUG initiation codon usage for DPR (poly-PG) synthesis may also apply to proteins translated from nucleotide repeat expansions in other genes. Lastly, our results support the idea that distinct translation initiation factors are involved in the synthesis of individual DPRs produced from the same nucleotide repeat expansion. Future studies focused on transcriptional and translational mechanisms of expanded nucleotide repeats may critically contribute to the design of therapies for these diseases.

## Materials and methods

**Key resources table**

| Reagent type (species) or resource | Designation | Source or reference | Identifiers | Additional information |
|---|---|---|---|---|
| Cell line (*Homo sapiens*) | HEK293 | ATCC | CRL-1573 | |
| Cell line (*Mus musculus*) | NSC34 | Gift from Dr. Neil R. Cashman (McGill University) PMID:1467557 | | |
| Cell line (*Homo sapiens*) | Isogenic iPS cells | *Lopez-Gonzalez et al., 2019* PMID:31019093 | 26z90 | Isogenic control for patient line C926#6 |
| Cell line (*Homo sapiens*) | Isogenic iPS cells | *Lopez-Gonzalez et al., 2019* PMID:31019093 | 27m91 | Isogenic control for patient line C927#11 |

*Continued on next page*

*Continued*

| Reagent type (species) or resource | Designation | Source or reference | Identifiers | Additional information |
|---|---|---|---|---|
| Cell line (*Homo sapiens*) | Healthy control iPS cells | *Almeida et al., 2013* PMID:23836290 | Control2#20 | Control for patient line C940#3 |
| Cell line (*Homo sapiens*) | C9orf72 patient iPS cells | *Almeida et al., 2013* PMID:23836290 | C926#6 | C9orf72 patient line |
| Cell line (*Homo sapiens*) | C9orf72 patient iPS cells | *Almeida et al., 2013* PMID:23836290 | C927#11 | C9orf72 patient line |
| Cell line (*Homo sapiens*) | C9orf72 patient iPS cells | *Freibaum et al., 2015* PMID:26308899 | C940#3 | C9orf72 patient line |
| Antibody | Anti-Poly-PR (Rabbit polyclonal) | EMD Millipore | ABN1354 | WB (1:1000) IF (1:250) |
| Antibody | Anti-Poly-PG (Mouse monoclonal) | Target ALS | TALS828.179 | WB (1:1000) IF (1:100) |
| Antibody | Anti-Poly-PA (Rabbit polyclonal) | EMD Millipore | ABN1356 | WB (1:1000) |
| Antibody | Anti-nLuc (Mouse monoclonal) | Promega | N700A | WB (1:500) |
| Antibody | Anti-α-tubulin (Rat monoclonal) | Abcam | Ab6160 | WB (1:5000) |
| Antibody | Anti-H3K4me2 (Rabbit polyclonal) | EMD Millipore | 07-030 | WB (1:2000) |
| Antibody | Anti-mouse horseradish peroxidase-conjugated secondary antibody (Sheep monoclonal) | GE Healthcare | NA931V | WB (1:5000) |
| Antibody | Anti-rabbit horseradish peroxidase-conjugated secondary antibody (Donkey monoclonal) | GE Healthcare | NA934V | WB (1:5000) |
| Antibody | Anti-rat horseradish peroxidase-conjugated secondary antibody (Goat polyclonal) | Cell Signaling Technology | 7077S | WB (1:1000) |
| Antibody | Alexa 488-conjugated anti-mouse IgG (Chicken polyclonal) | Thermo Fisher Scientific | A-21200 | IF (1:2000) |
| Antibody | Alexa 488-conjugated anti-rabbit IgG (Goat polyclonal) | Thermo Fisher Scientific | A-11008 | IF (1:2000) |
| Recombinant DNA reagent | pAG-ΔC9::nLuc | PMID:29792928 | | |
| Recombinant DNA reagent | pAG-AS(C9)-Poly-PR::nLuc (Plasmid) | This paper | | Plasmid vector containing 35 CCCCGG repeats preceded by 1000-bp-long intronic sequence from human *C9ORF72*, and NanoLuc in frame of poly-PR |
| Recombinant DNA reagent | pAG-AS(C9)-Poly-PG::nLuc (Plasmid) | This paper | | Plasmid vector containing 35 CCCCGG repeats preceded by 1000-bp-long intronic sequence from human *C9ORF72*, and NanoLuc in frame of poly-PG |
| Recombinant DNA reagent | pAG-AS(C9)-Poly-PA::nLuc (Plasmid) | This paper | | Plasmid vector containing 35 CCCCGG repeats preceded by 1000-bp-long intronic sequence from human *C9ORF72*, and NanoLuc in frame of poly-PA |

*Continued on next page*

*Continued*

| Reagent type (species) or resource | Designation | Source or reference | Identifiers | Additional information |
|---|---|---|---|---|
| Recombinant DNA reagent | pAG-AS(C9)$_{-366CUG->CCC}$-Poly-PR::nLuc (Plasmid) | This paper | | pAG-AS(C9)-Poly-PR::nLuc vector with mutation of the CTG at –366 bp from CCCCGG repeats to CCC |
| Recombinant DNA reagent | pAG-AS(C9)$_{-366CUG->UAG}$-Poly-PR::nLuc (Plasmid) | This paper | | pAG-AS(C9)-Poly-PR::nLuc vector with mutation of the CTG at –366 bp from CCCCGG repeats to TAG |
| Recombinant DNA reagent | pAG-AS(C9)$_{-273AUG->CCC}$-Poly-PR::nLuc (Plasmid) | This paper | | pAG-AS(C9)-Poly-PR::nLuc vector with mutation of the ATG at –273 bp from CCCCGG repeats to CCC |
| Recombinant DNA reagent | pAG-AS(C9)$_{-273AUG->UAG}$-Poly-PR::nLuc (Plasmid) | This paper | | pAG-AS(C9)-Poly-PR::nLuc vector with mutation of the ATG at –273 bp from CCCCGG repeats to TAG |
| Recombinant DNA reagent | pAG-AS(C9)$^{CCC}$-Poly-PG::nLuc (Plasmid) | This paper | | pAG-AS(C9)-Poly-PG::nLuc vector with mutation of ATG at –212 bp, ATG at –194 bp, CTG at -182 bp, and ATG at –113 bp from CCCCGG repeats to CCC |
| Recombinant DNA reagent | pAG-AS(C9)$_{-212AUG}$-Poly-PG::nLuc (Plasmid) | This paper | | pAG-AS(C9)-Poly-PG::nLuc vector with mutation of ATG at –194 bp, CTG at –182 bp, and ATG at –113 bp from CCCCGG repeats to CCC |
| Recombinant DNA reagent | pAG-AS(C9)$_{-194AUG}$-Poly-PG::nLuc (Plasmid) | This paper | | pAG-AS(C9)-Poly-PG::nLuc vector with mutation of ATG at –212 bp, CTG at –182 bp, and ATG at –113 bp from CCCCGG repeats to CCC |
| Recombinant DNA reagent | pAG-AS(C9)$_{-182CUG}$-Poly-PG::nLuc (Plasmid) | This paper | | pAG-AS(C9)-Poly-PG::nLuc vector with mutation of ATG at –212 bp, ATG at –194 bp, and ATG at –113 bp from CCCCGG repeats to CCC |
| Recombinant DNA reagent | pAG-AS(C9)$_{-113AUG}$-Poly-PG::nLuc (Plasmid) | This paper | | pAG-AS(C9)-Poly-PG::nLuc vector with mutation of ATG at –212 bp, ATG at –194 bp, and CTG at –182 bp from CCCCGG repeats to CCC |
| Recombinant DNA reagent | pAG-AS(C9)$_{-212CCC}$-Poly-PG::nLuc (Plasmid) | This paper | | pAG-AS(C9)-Poly-PG::nLuc vector with mutation of ATG at –212 bp from CCCCGG repeats to CCC |
| Recombinant DNA reagent | pAG-AS(C9)$_{-194CCC}$-Poly-PG::nLuc (Plasmid) | This paper | | pAG-AS(C9)-Poly-PG::nLuc vector with mutation of ATG at –194 bp from CCCCGG repeats to CCC |
| Recombinant DNA reagent | pAG-AS(C9)$_{-113CCC}$-Poly-PG::nLuc (Plasmid) | This paper | | pAG-AS(C9)-Poly-PG::nLuc vector with mutation of ATG at –113 bp from CCCCGG repeats to CCC |
| Recombinant DNA reagent | pAG-AS(C9)$_{-212UAG}$-Poly-PG::nLuc (Plasmid) | This paper | | pAG-AS(C9)-Poly-PG::nLuc vector with mutation of ATG at –212 bp from CCCCGG repeats to TAG |
| Recombinant DNA reagent | pAG-AS(C9)$_{-194UAG}$-Poly-PG::nLuc | This paper | | pAG-AS(C9)-Poly-PG::nLuc vector with mutation of ATG at –194 bp from CCCCGG repeats to TAG |
| Recombinant DNA reagent | pAG-AS(C9)$_{-113UAG}$-Poly-PG::nLuc (Plasmid) | This paper | | pAG-AS(C9)-Poly-PG::nLuc vector with mutation of ATG at –113 bp from CCCCGG repeats to TAG |
| Recombinant DNA reagent | lentiCRISPR v2-*EIF2D* (Plasmid) | This paper | Addgene (#52961) | lentiCRISPR plasmid containing gRNA sequence against *EIF2D* |

*Continued on next page*

*Continued*

| Reagent type (species) or resource | Designation | Source or reference | Identifiers | Additional information |
|---|---|---|---|---|
| Recombinant DNA reagent | Sh-Control (Plasmid) | PMID:34654821 | Thermo Fisher Scientific (#AM5764) | pSilencer 2.1-U6 neo plasmid containing non-specific control shRNA sequence |
| Recombinant DNA reagent | Sh-*EIF2D* (Plasmid) | PMID:34654821 | Thermo Fisher Scientific (#AM5764) | pSilencer 2.1-U6 neo plasmid containing shRNA sequence against *EIF2D* |
| Recombinant DNA reagent | pGL4.50 [luc2/CMV/Hygro] (Plasmid) | Promega | E131A | Expression of firefly luciferase |
| Recombinant DNA reagent | pNL1.1 CMV (Plasmid) | Promega | N109A | Expression of NanoLuc |
| Recombinant DNA reagent | pcDNA 6/V5-His A (Plasmid) | Thermo Fisher Scientific | 43-0003 | |
| Sequence-based reagent | siRNA: non-targeting negative control | Thermo Fisher Scientific | 4390844 | Silencer Select |
| Sequence-based reagent | siRNA: *EIF2D* | Thermo Fisher Scientific | S4495 | Silencer Select |
| Sequence-based reagent | siRNA: *EIF2D* | Thermo Fisher Scientific | S4496 | Silencer Select |
| Chemical compound, drug | Halt Protease Inhibitor Cocktail | Thermo Fisher Scientific | 87786 | |
| Chemical compound, drug | SB421542 | Stemgent | 04-0010-10 | Neuron differentiation |
| Chemical compound, drug | CHIR99021 | Stem Cell Technologies | 72054 | Neuron differentiation |
| Chemical compound, drug | DMH1 | Stem Cell Technologies | 73634 | Neuron differentiation |
| Chemical compound, drug | All-Trans Retinoic Acid | Stem Cell Technologies | 72262 | Neuron differentiation |
| Commercial assay or kit | Q5 Site-Directed Mutagenesis Kit | New England Biolabs | E0554S | |
| Commercial assay or kit | Nano-Glo Dual-Luciferase Reporter assay system | Promega | N1610 | |
| Commercial assay or kit | Cell Counting Kit-8 | Dojindo | CK-04 | |
| Commercial assay or kit | BCA Protein Assay Kit | Thermo Fisher Scientific | 23225 | |
| Commercial assay or kit | 660 nm Protein Assay Reagent | Thermo Fisher Scientific | 22660 | |
| Software, algorithm | Image Lab software | Bio-Rad | | |
| Software, algorithm | ImageJ2 software | PMID:22930834 | | |
| Software, algorithm | GraphPad Prism | Dotmatics | | |
| Other | 5× passive lysis buffer | Promega | E1941 | Lysis buffer for luciferase assay |
| Other | 4',6-diamidino-2-phenylindole (DAPI) | Thermo Fisher Scientific | D1306 | Nuclear staining (1 mg/ml) |
| Other | SuperSignal West Dura Extended Duration Substrate | Thermo Fisher Scientific | 34076 | Horseradish peroxidase substrate for western blotting |
| Other | Lipofectamine LTX | Thermo Fisher Scientific | 15338030 | Plasmid transfection reagent |

## Generation of the plasmid constructs

All oligonucleotides were obtained from Integrated DNA Technologies. Oligonucleotide I-F/R (*Supplementary file 1*) contains part of a *Hin*dIII site followed by 113 nucleotides that are normally upstream

of the GGGGCC repeats and then by three GGGGCC repeats. Oligonucleotide II-F/R contains 10 GGGGCC repeats followed by part of a *Not*I site. These two oligonucleotides were phosphorylated, annealed, and then ligated into restriction sites of *Hind*III and *Not*I of a pAG plasmid. The plasmid was then digested with HindIII and *Bam*HI. The *Hind*III-*Bam*HI fragment was digested with *Ban*II, and the resultant *Hind*III-*Ban*II fragment was then ligated with oligonucleotide II-F/R into the pAG plasmid. This approach was repeated three times with similar digestions and ligations of oligonucle-otide II. Finally, the *Hind*III-*Ban*II fragment was ligated with oligonucleotide III-F/R (which contains two CCCCGG repeats followed by a 99 bp flanking sequence and then followed by part of the *Not*I site) into the pAG plasmid (referred to as 113bp-35RG4C2-99bp plasmid). To delete stop codons after the CCCCGG repeats, the plasmid was treated with Bfal and NotI, and the digested fragment was ligated with oligonucleotide IV-F/R. To add sequence upstream from the C4G2 repeats, a 543 bp portion (408–950 of NCBI reference sequence, NC_000009.12) of the *C9ORF72* gene from HEK293 genomic DNA was amplified by PCR using the primer shown in *Supplementary file 1*. The amplified construct was then ligated with the BtgI/NotI-digested fragment of the 113bp-35RG4C2-99bp plasmid into XbaI and NotI sites of pcDNA6/V5-His A plasmid (referred to as 609bp-35RC4G2 plasmid). To further increase the length of sequence upstream from CCCCGG repeats, a 392 bp portion (951-1342 of NCBI reference sequence, NC_000009.12) of *C9ORF72* gene from HEK293 genomic DNA was amplified by PCR using the primer shown in *Supplementary file 1*. The amplified construct was then ligated with the XbaI/NotI fragment of 609bp-35RC4G2 plasmid into HindIII and NotI sites of the pAG plasmid (referred to as AS-C9 plasmid). The ΔC9 plasmid (*Sonobe et al., 2021*) was generated as previously described.

To mutate sequences, a 560 bp portion upstream from the repeats in the AS-C9 plasmid was amplified by PCR using a primer shown in *Supplementary file 1*. The amplified portion was then ligated into the HindIII and NotI sites of pcDNA6/V5-His A plasmid. Mutations were made with Q5 Site-Directed Mutagenesis Kit (New England Biolabs) using primer sets (*Supplementary file 1*). The StuI/BtgI portion of the resultant mutants was then cloned back into the StuI and NotI sites of AS-C9 plasmid with BtgI/NotI portion of AS-C9 plasmid using the primer sets in *Supplementary file 1*.

To generate the vector to induce expression of poly-PA, the fragment AUG-PA-F/R (*Supplementary file 1*) was phosphorylated, annealed, and then ligated into restriction sites of HindIII and BtgI of the AS-C9 plasmid.

## Cell culture

HEK293 and NSC34 cells were cultured in DMEM supplemented with 10% FBS, 2 mM L-glutamine, 100 U/ml penicillin, and 100 µg/ml streptomycin. The cell lines were checked for mycoplasma contam-ination by DAPI staining but were not authenticated.

## Luciferase assay

The cells were plated in 24-well plates at $5\times10^4$ cells per well and then cotransfected using Lipofect-amine LTX (Thermo Fisher Scientific) with 100 ng of the plasmid along with 100 ng fLuc plasmid as a transfection control. After 48 hr, the cells were lysed with 1× passive lysis buffer (Promega). Levels of nLuc and fLuc were assessed with the Nano-Glo Dual-Luciferase Reporter assay system (Promega) and a Wallac 1420 VICTOR 3V luminometer (Perkin Elmer) according to the manufacturer's protocol.

## Western blotting

The cells were plated in six-well plates at $2\times10^5$ cells per well and then cotransfected with 2.5 µg of plasmids using Lipofectamine LTX (Thermo Fisher Scientific). After 48 hr, cell lysates were prepared using RIPA buffer (50 mM Tris-HCl, pH 7.5; 150 mM NaCl; 0.1% SDS; 0.5% sodium deoxycholate; 5 mM EDTA containing 1× Halt Protease inhibitor Cocktail). The RIPA-insoluble pellet was lysed in 8 M urea and used as the RIPA-insoluble fraction. H3K4me2 was used as marker for RIPA-insoluble fraction, as previously described (*Janes, 2015*). Lysates were subjected to electrophoresis on Mini-PROTEAN TGX Gels (Bio-Rad), and then transferred to Amersham Hybond P 0.45 µm PVDF membranes (GE Health-care). The membrane was blocked with 5% non-fat skim milk in Tris-buffered saline containing 0.05% Tween-20 for 1 hr at room temperature, and then incubated overnight at 4°C with primary antibodies against poly-PR (1:1000, ABN1354, EMD Millipore), poly-GP (1:1000, TALS 828.179, Target ALS), eIF2D (1:1000, 12840-1-AP, Proteintech), poly-PA (1:1000, ABN1356, EMD Millipore), nLuc (1:500,

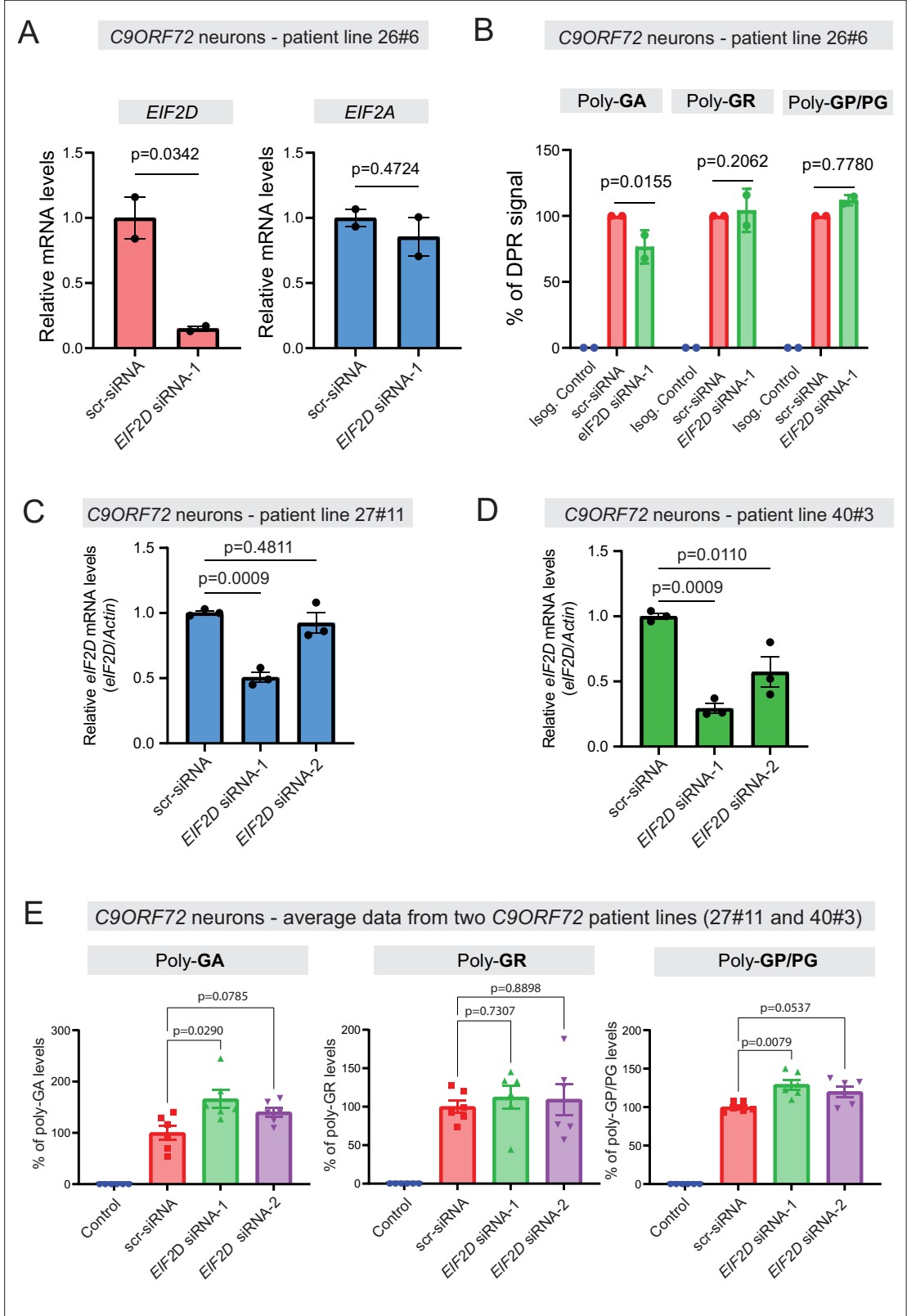

**Figure 7.** Dipeptide repeat (DPR) levels in human iPSC-derived neurons upon eIF2D knockdown. (**A**) The *EIF2D*, *EIF2A*, and *actin* mRNA levels were assessed by real-time quantitative PCR on either isogenic control (26Z90) or *C9ORF72* human motor neurons (patient line 26#6) upon small interfering RNA (siRNA) transfection (scramble or EIF2D siRNA-1). The *eIF2D* and *eIF2A* mRNA levels were normalized to actin. The experiments were repeated twice. p<0.05 by one-way ANOVA with Tukey's post hoc test. (**B**) Poly-GA, poly-GR, and poly-GP levels in motor neurons differentiated independently

*Figure 7 continued on next page*

*Figure 7 continued*

(twice) from isogenic control and one *C9ORF72* iPSC line. DPR levels were measured using an Meso Scale Discovery (MSD) immunoassay in a blinded manner. Data presented as mean ± SD. p-Values were calculated using two-way ANOVA with Dunnett's multiple comparison test using Prism (9.1) software. (**C–D**) The *EIF2D* and *actin* mRNA levels were assessed by real-time quantitative PCR on *C9ORF72* human motor neurons (two patient lines) upon siRNAs transfection (scramble, EIF2D siRNA-1 or EIF2D siRNA-2). The *eIF2D* mRNA levels were normalized to actin. The experiments were repeated three times. *p<0.05, ***p<0.001, ns, not significant by two-tailed unpaired t tests were used for two groups and a one-way ANOVA followed by Dunnett's post hoc analysis was used for more than two groups. (**E**) Poly-GA, poly-GR, and poly-GP levels in motor neurons differentiated independently (n=3 times) from isogenic or healthy control lines and total two *C9ORF72* patient iPSC lines (lines 27#11 and 40#3). DPR levels were measured using an MSD immunoassay in a blinded manner. For poly(GA) assay, total protein normalized poly(GA) concentrations were converted to percentage and presented as mean ± SE. For poly(GR), poly(GP) assay, total protein normalized electrochemiluminescence (ECL) values were converted to percentage and presented as mean ± SE. p-Values were calculated using one-way ANOVA with Dunnett's T3 multiple comparisons test .

The online version of this article includes the following source data and figure supplement(s) for figure 7:

**Figure supplement 1.** The small interfering RNA (siRNA) against eIF2D knocks down eIF2D protein levels.

**Figure supplement 1—source data 1.** Full raw unedited images of western blots shown in *Figure 7—figure supplement 1*.

---

N700A, Promega), α-tubulin (1:5000, YL1/2, Abcam), and dimethyl-histone H3 (H3K4me2) (1:2000, 07-030, EMD Millipore). Following washing, the membrane was incubated for 1 hr at room temperature with anti-mouse (1:5000, GE Healthcare), anti-rabbit (1:5000, GE Healthcare), or anti-rat horseradish peroxidase-conjugated secondary antibodies (1:1000, Cell Signaling Technology). The signal was detected using SuperSignal West Dura Extended Duration Substrate (Thermo Fisher Scientific) and analyzed using ChemiDoc MP Imaging System and Image Lab software (version 6.0.1, Bio-Rad).

## Cell viability assay

Cell viability assay was performed using Cell counting kit-8 (Dojindo) according to the manufacturer's protocol. In brief, NSC34 cells were plated in 96-well plates at $2.5 \times 10^3$ cells per well and then transfected using Lipofectamine LTX with 100 ng of the indicated plasmid. After 48 hr, 10 μl of the CCK-8 solution was added to the well and incubated for 2 hr in a $CO_2$ incubator. The reaction was stopped by adding 0.1 M HCl and the absorbance at 450 nm was measured.

## Immunocytochemistry

The cells were plated in four-well Lab-Tek II Chamber Slide (Nunc) coated with 50 μg/ml poly-D-lysine (Sigma) at $5 \times 10^4$ cells per well and transfected using Lipofectamine LTX with 500 ng of the indicated plasmid. After 48 hr, the cells were fixed with 4% paraformaldehyde for 15 min at room temperature. Then, the cells were permeabilized with phosphate buffered saline (PBS) with 0.2% Tween-20 for 20 min at room temperature. The samples were incubated with blocking buffer (2% BSA in PBS) for 1 hr at room temperature and then incubated overnight at 4°C with antibodies against poly-PR (1:250, ABN1354, EMD Millipore) or poly-GP (1:100, TALS 828.179, Target ALS). After rinsing with PBS, cells were incubated with Alexa 488-conjugated chicken anti-mouse IgG (1:2000, Thermo Fisher Scientific) or Alexa 488-conjugated goat anti-rabbit IgG (1:2000, Thermo Fisher Scientific) for 1 hr at room temperature, and then counterstained with DAPI. Images were captured using a confocal laser microscope system (Leica TCS SP5, Leica Microsystems) and processed using ImageJ2 software (version 2.9.0/1.53t).

## Generation of *EIF2D* knockout cells by CRISPR/Cas9 gene editing

A single guide RNA (sgRNA) (GCAGTGACTGTGTACGTGAG) that targets exon 2 of eIF2D was cloned into lentiCRISPR v2 plasmid (Addgene). HEK293 cells were plated into six-well plates at $4 \times 10^5$ cells per well, and then transfected using Lipofectamine LTX with 2.5 μg lentiCRISPR v2 plasmids containing the sgRNA sequence. Transfected cells were selected using 3 μg/ml puromycin for 3 days. *EIF2D* knockout cell clones were obtained by limited dilution. The resulting *EIF2D* knockout cells carry allele-specific mutations, as follows. Compared to the wild type (WT) GGATGCAGTGACTGTGTACGTGAG TGGTGG sequence, one allele GGATGCAGTGACTGTGTACG**T**TGAGTGGTGG has a single nucleotide insertion shown bolded while the other allele contains a two-nucleotide deletion GGATGCAG TGACTGTGTA—TGAGTGGTGG. Both alleles lead to a premature stop codon, likely resulting in two different truncated eIF2D proteins with the following respective sequence:

MFAKAFRVKSNTAIKGSDRRKLRADVTTAFPTLGTDQVSELVPGKEELNIVKLYAHKGDAVTVYEWW and MFAKAFRVKSNTAIKGSDRRKLRADVTTAFPTLGTDQVSELVPGKEELNIVKLY AHKGDAVTVYVEWW.

## Knockdown of eIF2D in HEK293 cells

shRNA plasmids against human eIF2D were prepared using previously published methods (*Sonobe et al., 2021*). In brief, oligonucleotides with an siRNA sequence were cloned into the *Bam*HI and *Hin*dIII sites of p*Silencer* 2.1-U6 neo Vector (Thermo Fisher Scientific) according to the manufacturer's protocol. The latter kit also contained a control shRNA vector. For luciferase assays (shown above), the cells were plated in 24-well plates at $5 \times 10^4$ cells per well and cotransfected with 50 ng of the AS-C9 plasmids and 50 ng of the fLuc plasmids along with 500 ng of either control shRNA or anti-eIF2D shRNA using Lipofectamine LTX (Thermo Fisher Scientific).

## Motor neuron differentiation from human iPSC lines

Human motor neurons were differentiated as previously described from a published iPSC line obtained from a *C9ORF72* carrier (FTD26-6), as well as an isogenic control line that had a CRISPR/Cas9-mediated deletion of expanded GGGGCC repeats (*Lopez-Gonzalez et al., 2019*; *Lopez-Gonzalez et al., 2016*). Briefly, iPSCs were plated and expanded in mTSER1 medium (Stem Cell Technologies) in Matrigel-coated wells. Twenty-four hours after plating, the culture medium was replaced every other day with neuroepithelial progenitor (NEP) medium, DMEM/F12 (Gibco), neurobasal medium (Gibco) at 1:1, 0.5X N2 (Gibco), 0.5X B27 (Gibco), 0.1 mM ascorbic acid (Sigma), 1X Glutamax (Invitrogen), 3 µM CHIR99021 (Tocris Bioscience), 2 µM DMH1 (Tocris Bioscience), and 2 µM SB431542 (Stemgent) for 6 days. NEPs were dissociated with accutase, split 1:6 into Matrigel-coated wells, and then cultured for 6 days in motor neuron progenitor induction medium (NEP with 0.1 µM retinoic acid and 0.5 µM purmorphamine, both from Stemgent). Motor neuron progenitors were dissociated with accutase to generate suspension cultures, and the cells were cultured in motor neuron differentiation medium (NEP with 0.5 µM retinoic acid and 0.1 µM purmorphamine). After 6 days, the cultures were dissociated into single cells, and seeded on Matrigel-coated plates in motor neuron medium, 0.5X B27 supplement, 0.1 mM ascorbic acid, 1X Glutamax, 0.1 µM Compound E (Calbiochem), 0.26 µg/ml cAMP, 1 µg/ml Laminin (Sigma), 10 ng/ml GDNF (R&D Systems), and 10 ng/ml GDNF (R&D Systems), and 10 ng/ml BDNF. Motor neurons were cultured for 5 weeks.

## SiRNA knockdown

After 3 weeks in neuron culture media, motor neurons were transfected with an siRNA specific to *eIF2D* mRNA or a scrambled control. For the transfection, Lipofectamine RNAiMAX (Thermo Fisher Scientific) was first diluted in Opti-MEM medium, and then both eIF2D and scrambled control siRNAs were separately diluted in Opti-MEM medium at room temperature. Diluted siRNA and diluted Lipofectamine RNAiMAX (1:1 ratio) were then mixed and incubated for 20 min. The siRNA-lipid complex solution was then brought up to the appropriate volume with MN culture medium. The culture medium in the plate was aspirated and replaced with an siRNA-lipid complex at a final concentration of 60 pmol siRNA in 1.5 ml medium per 1,000,000 cells. After 24 hr, the medium was replaced with a normal motor neuron medium. This process was repeated two more times at 26 and 31 days in culture. After 36 days in culture, we measured siRNA efficiency and levels of DPRs in harvested motor neurons.

## RNA extraction and quantitative real-time PCR

Total RNA from iPSC-derived motor neurons was extracted with the RNeasy Mini Kit (QIAGEN) and then reverse-transcribed to cDNA with the TaqMan Reverse Transcription Kit (Applied Biosystems). Quantitative PCR was carried out with SYBR Green Master Mix (Applied Biosystems). Using primers listed in SI Appendix, Table, Ct values for each gene were normalized to actin and GAPDH. Relative mRNA expression was calculated with the double delta Ct method.

## Measurement of soluble poly-GR and poly-GP in iPSC-derived neurons

Soluble poly-GR and poly-GP levels in iPSC-derived neurons were detected using the Meso Scale Discovery (MSD) Immunoassay platform as previously reported (*Krishnan et al., 2022*). In brief, cells were lysed using Tris-based lysis buffer, and lysates were adjusted to equal concentrations and loaded

in duplicate wells. Background subtracted electrochemiluminescence signals were presented as percentage. The MSD assays were performed in a blinded manner.

## Soluble and insoluble fractionation for measurement of poly-GA

Motor neurons were lysed in RIPA buffer (Boston BioProducts, BP-115D) with protease and phosphatase inhibitors. The lysates were rotated for 30 min at 4°C, followed by centrifugation at 13,500 rpm for 20 min. The supernatant was removed and used as the soluble fraction. Protein concentrations of the soluble fraction were determined by the BCA assay (Thermo Fisher Scientific, Cat # 23227). To remove carryovers, the pellets were washed with RIPA buffer, and then resuspended in the same buffer with 2% SDS followed by sonication on ice. The lysates were rotated for 30 min at 4°C, then spun at 14,800 rpm for 20 min at 4°C. The supernatant was removed and used as insoluble fraction. Protein concentrations of the insoluble fraction were determined by Pierce 660 nm Protein Assay (Thermo Fisher Scientific, 22660).

## Measurement of poly-GA in iPSC-derived neurons

Poly-GA in soluble motor neuron lysates was measured using an MSD sandwich immunoassay. A human/murine chimeric form of anti-GA antibody chGA3 was used for capture, and a human anti-GA antibody GA4 with a SULFO-tagged anti-human secondary antibody was used for detection. Poly-GA concentrations were interpolated from the standard curve using 60X-GA expressed in HEK 293 cells and presented as percentage. For background correction, values from no-repeats neuron samples were subtracted from the corresponding test samples.

## Statistical analysis

Statistical analysis was performed by one-way ANOVA with Tukey's multiple comparison test and two-way ANOVA with the Šídák multiple comparison test using GraphPad Prism version 9.3.1. A p-value of $<0.05$ was considered significant. The data are presented as mean $\pm$ standard error of the mean.

## Acknowledgements

This work was supported by a grant from the Lohengrin Foundation (RPR, PK), a basic science pilot grant from the Association for Frontotemporal Degeneration (AFTD) (RPR, PK), and two NIH grants (R37NS057553 and R01NS101986) to FBG. The antibodies used to measure GA levels were discovered by Neurimmmune AG (Zurich, Switzerland).

---

# Additional information

### Competing interests

Yuanzheng Gu, Deborah Y Kwon: affiliated with Biogen. The authors have no financial interests to declare. Paschalis Kratsios: Reviewing editor, *eLife*. The other authors declare that no competing interests exist.

### Funding

| Funder | Grant reference number | Author |
| --- | --- | --- |
| Association for Frontotemporal Degeneration | | Paschalis Kratsios |
| National Institute of Neurological Disorders and Stroke | R37NS057553 | Fen-Biao Gao |
| National Institute of Neurological Disorders and Stroke | R01NS101986 | Fen-Biao Gao |

| Funder | Grant reference number | Author |
|--------|------------------------|--------|

The funders had no role in study design, data collection and interpretation, or the decision to submit the work for publication.

## Author contributions

Yoshifumi Sonobe, Conceptualization, Data curation, Formal analysis, Validation, Investigation, Visualization, Methodology; Soojin Lee, Gopinath Krishnan, Yuanzheng Gu, Formal analysis, Investigation, Methodology; Deborah Y Kwon, Formal analysis, Investigation; Fen-Biao Gao, Formal analysis, Supervision, Funding acquisition, Project administration, Writing – review and editing; Raymond P Roos, Conceptualization, Formal analysis, Supervision, Funding acquisition, Writing – original draft, Project administration, Writing – review and editing; Paschalis Kratsios, Conceptualization, Formal analysis, Supervision, Funding acquisition, Investigation, Visualization, Writing – original draft, Project administration, Writing – review and editing

## Author ORCIDs

Paschalis Kratsios ⓘ http://orcid.org/0000-0002-1363-9271

## Decision letter and Author response

Decision letter https://doi.org/10.7554/eLife.83189.sa1
Author response https://doi.org/10.7554/eLife.83189.sa2

## Additional files

### Supplementary files
- Supplementary file 1. List of primers used for this study.
- Transparent reporting form

### 1Data availability

All data generated or analyzed during this study are included in the manuscript and supporting files.

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
