## [Editor Report]

This study by Sonobe et al. uses transfected cells and patient iPSC-derived neurons to define mechanisms underlying translation of the antisense CCCCGG RNA strand expressed in *C9ORF72*-associated ALS and FTD. The authors design a series of constructs to explore the start codon required to produce toxic PR and prominent PG dipeptides in disease. Using these constructs they provide solid data that translation in the PR and PG reading frames occurs due to the presence of AUG codons within the 5'UTR of the RNA strand.

---

## [Decision Letter]

**Decision letter after peer review:**

Thank you for submitting your article "Translation of dipeptide repeat proteins in C9ORF72 ALS/FTD through unique and redundant AUG initiation codons" for consideration by *eLife*. Your article has been reviewed by 2 peer reviewers, and the evaluation has been overseen by a Reviewing Editor and Jeannie Chin as the Senior Editor. The reviewers have opted to remain anonymous.

Essential revisions:

1) Further validation of data in Figure 5 as per comments from Reviewers.

2) Address the contribution of sense-strand encoded GP versus antisense-strand encoded PG in iPSC studies as per Reviewer comments.

3) All western immunoblot data need to be quantified and statistically analyzed as outlined by Reviewer #1.

4) The inclusion of additional iPSN lines to demonstrate findings are consistent within an endogenous setting as outlined by Reviewer #2.

5) Further validation of the luciferase assay to address concerns raised by Reviewers.

6) An evaluation of cited studies to confirm accuracy of reported information as per concerns raised by Reviewer #1.

7) An evaluation of DPR aggregation to demonstrate relevance in disease as outlined by Reviewer #1.

*Reviewer #1 (Recommendations for the authors):*

1. All western immunoblot data need to be quantified and statistically analyzed. In particular, quantification of the multiple molecular weight bands in Figures 2B, 3B and 4B is needed to support the authors conclusions.

2. There are several issues with the data in figure 5.

a. the successful knockdown of eIF2D needs to be confirmed on the protein level.

b. multiple siRNA targeting eIF2D should be tested to confirm that effects are specific to the targeted gene versus an off-target effect.

c. the authors need to be clearer about their interpretation of the data on GP/PG at this point in their study. Specifically, the GP that is measured here will include both sense-GP and antisense-PG. While Zu et al. demonstrated that the antisense strand is the primary source of GP/PG (which the authors should mention in the intro or results for the reader) the sense strand will still contribute to DPR production and could mask effects. In Figure 6D & F, the authors present more compelling data to support that eIF2D is dispensable for PG production. The authors may want to consider presenting those data sooner.

3. The authors design novel C4G2:NanoLuc constructs to study antisense DPR synthesis. Further validation of the constructs and the system would improve the current study.

a. Can the authors present data showing that the PR and PG bands in Figure 1D-1E co-stain with an anti-luciferase antibody since the DPR are tagged with nLuc (Figure 1A)? This would demonstrate that the luciferase signal is an effective and accurate readout for DPR production.

b. The authors should consider insoluble versus soluble DPR when measuring DPR production. They only look at soluble DPR but the DPR levels within the insoluble fraction may differ (Quaegebeur et al. 2020; Licata et al. 2021) and this may impact the interpretation of the later data as the DPR may be enriched in the insoluble fraction and seem depleted in the soluble fraction measured using western immunoblots and the luciferase assay. Thus, can the authors demonstrate that the levels of DPR in the insoluble fraction are consistent with the soluble fraction in their system making the soluble fraction a reasonable read-out?

c. It is surprising that the authors do not see PA production in Figure 1 using their luciferase assay. Can the authors confirm these results using an antibody against PA? The authors should discuss why they do not see any PA production when PA is seen in patient samples.

4. The authors should consider changes in cellular toxicity in their experiments as they are discussing potential impacts in human disease. Can they demonstrate that their manipulations alter cell survival or DPR-associated toxicity? what about changes in DPR aggregation (since GP/PG may not be toxic but these do aggregate and this is pathologically relevant to patients)?

5. The authors need to carefully check their citations and make sure the cited studies are accurately being described. Here are some examples of issues

a. Lines 56, 202-203 and throughout: the statements "… is independent of G4C2 repeats" don't make sense. The G4C2 repeat is what codes for the DPR so DPR production cannot be independent of the G4C2 sequence. Please rephrase these sentences to better describe previous findings that translation is mediated by flanking sequence. Also, the authors may want to be more reflective when discussing this point. Specifically, the presence of an expanded G4C2 repeat alone without flanking sequence can result in DPR production in cultured cells and in flies. So while translation may be promoted by flanking sequence in the cited manuscripts, translation may also occur through RAN translation mechanisms that solely utilize the repeat. Both modes of translation may occur simultaneously in disease while the use of non-canonical and canonical start codons may be the primary contributor of DPR production.

b. Line 224: the sentence describing Goodman et al. is incorrect. This study utilized a fly model that carried a ~100bp of the 5' flanking sequence found in patients and thus was not a G4C2 repeat alone as described. Further, eIF4B/H was not show to "directly bind" the G4C2 RNA but rather there was a correlation in eIF4B/H downregulation and reduced GR-GFP production with no impact on non-repeat transcripts.

c. Line 42: it may be more transparent to change "thought to" to "predicted to" for readers unfamiliar with the topic. "Thought to" indicates that there is evidence to support all 3 potential modes of toxicity vs predicted to which clearly indicates these are hypothetical scenarios. However, the contribution of reduced C9orf72 protein to disease is not well established. Further, this point needs to be de-emphasized in favor of gain-of-function modes of toxicity by listing it as the last potentiality versus the first potentiality.

d. Line 68: this is not an exhaustive list of RAN translation factors identified for G4C2. There is also DHX36 (Tseng et al. 2021) and eIF4B/eIF4H (Goodman et al. 2019; supported by data in Linsalata et al. 2019). There may be others as well. Please add missing information.

e. Please add citations after the following statements and check for additional instances where citations are missing.

i. Line 43: citations for loss of C9orf72 potentially contributing to disease

ii. Line 52: citations for CUG for GA and AGG for GR.

iii. Line 58: citation for PG synthesis using AUG

iv. Line 63: citation for the failed clinical trial

*Reviewer #2 (Recommendations for the authors):*

In all the luciferase reporter figures, the relative values of all the constructs should be provided. It is expected that mutating AUGs will drastically reduce the translation level. However, whether the mutant reporter still have higher luciferase (by low efficient RAN translation) than no repeat control was not shown. This information needs to be included.

It has not been resolved what is the contribution of GP from sense and antisense strands. It has been shown in previous publications that reduction of sense repeats can significantly decrease the GP level. Therefore, there is not sufficient evidence to claim that knocking down eIF2D had no effect on GP level is due to the AUG translation of antisense repeats.

Only one isogeneic iPSC line was used in the study. 3-5 individual lines should be used to demonstrate the conclusion.

To prove the endogenous DPR production from the antisense repeats starts from AUG, the regions containing the AUG codons should be mutated/deleted to assess how it affects the DPR level.

In Figure 5, eIF2D protein level should be measured by western blotting.

In figure 6, the GA level should be measured as the positive control.

[Editors’ note: what follows is the authors’ response to the second round of review.]

Thank you for resubmitting your work entitled "Translation of dipeptide repeat proteins in *C9ORF72* ALS/FTD through unique and redundant AUG initiation codons" for further consideration by *eLife*. Your revised article has been evaluated by Carlos Isales (Senior Editor) and a Reviewing Editor.

The manuscript has been improved but there are some remaining issues that need to be addressed, as outlined below:

Edits are needed to address concerns raised by Reviewer #3. In particular, the new data on eIF2D as a GA-specific RAN translation factor has lead to some conflicting results with previous studies. This conclusion needs to be more fairly presented in the text, especially in the abstract and discussion.

*Reviewer #3 (Recommendations for the authors):*

The authors included new data to address most of the concerns and significantly improved the manuscript. However, as the reviewer was concerned, some of the conclusions cannot be supported when tested in more patient lines.

In the two additional patient iPSN lines, eIF2D knockdown did not reduce the GA level, but instead increased the GA level. Although the authors reasoned that the siRNA did not work well in the other two lines, the line 40#3 has good knockdown, and the upregulation of GA in two out of three lines is hard to interpret. Furthermore, the data in Figure 6 also did not show a big effect of reducing eIF2D (both KO and shRNA) on poly-GA level. Overall, the data cannot support the claim that eIF2D is generally required for poly-GA translation from a CUG initiation codon.

For the antisense repeat translation, the data in the manuscript showed the AUG can be used if the transcription starts more upstream. But without data from endogenous mutagenesis, this cannot be validated. The writing needs to be modified to clarify this point.

There is no Figure 7—figure supplement 1.

---

## [Author Response]

Essential revisions:1) Further validation of data in Figure 5 as per comments from Reviewers.

We have now tested a second siRNA against eIF2D and results are included in Figure 7.

We observed efficient eIF2D knockdown at the protein level with Western blotting upon transfection with one siRNA against eIF2D (Figure 7 —figure supplement 1). This experiment was conducted in HEK293 cells. The exact same siRNA (EIF2D-siRNA-1) was transfected in the iPSC-derived motor neurons from C9ORF72 patients (Figure 7). In iPSC-derived motor neurons, we assessed the efficiency of *EIF2D* knockdown at the mRNA level. Unfortunately, we were not able to conduct the experiment and measure eIF2D protein levels upon siRNA transfection because we used the entire amount of the lysates for DPR detection by immunoassays.

2) Address the contribution of sense-strand encoded GP versus antisense-strand encoded PG in iPSC studies as per Reviewer comments.

To specifically address this issue, the lab of one of the authors (Dr. Fen-Biao Gao) is currently preparing a separate manuscript. Specifically, the Gao lab tried multiple times to mutate the AUG codons in the poly-PG frame with CRISPR/Cas9 gene editing, but the region is very rich in GC content and the approach was unsuccessful. In this upcoming separate manuscript by the Gao lab, the authors were successful in generating a deletion of the intronic region that contains multiple AUG codons.

3) All western immunoblot data need to be quantified and statistically analyzed as outlined by Reviewer #1.

We have now performed these quantifications and statistical comparisons, which can be found in Figure 3 —figure supplement 1, Figure 4 —figure supplement 1, and Figure 5 —figure supplement 1.

4) The inclusion of additional iPSN lines to demonstrate findings are consistent within an endogenous setting as outlined by Reviewer #2.

We conducted new experiments. To test the role of eIF2D, we have now used three published iPSC lines from *C9ORF72* carriers, as well as their isogenic control lines which had CRISPR/Cas9-mediated deletion of expanded GGGGCC repeats. As we describe in Results, we only achieved (~90%) efficient EIF2D knockdown in neurons derived from one iPSC line from a *C9ORF72* carrier. In the remaining two lines, we witnessed variability in our siRNA approach to knockdown EIF2D. We now include these new data in Figure 7, but refrain from drawing strong conclusions on the role of EIF2D in DPR synthesis in the context of neurons derived from iPSCs of *C9ORF72* carriers. Hence, we describe in detail our iPSC experiments in Results, but we do not make any strong claims in the Abstract and Discussion.

On the other hand, our findings on the identification of AUG initiation codons are consistent with an endogenous setting. Zu et al., (PNAS, 2013) conducted 5’ Rapid Amplification of cDNA Ends (RACE) on brain samples of *C9ORF72* ALS/FTD patients. Although this analysis did not identify the exact transcription start site for the antisense CCCCGG RNA, it did show that the region that includes the AUG codons, which we found to be important for poly-PR or poly-PG, is included in the antisense RNA from human *C9ORF72* ALS/FTD samples.

We have modified the Results (lines 133-136) to: “These results strongly suggest that AUG at -273 bp is the start codon for translation of poly-PR, one of the most toxic DPRs in C9ORF72 ALS/FTD. This AUG is predicted to be included in the endogenous antisense CCCCGG transcript based on 5’ Rapid Amplification of cDNA Ends (RACE) analysis on brain samples of C9ORF72 ALS/FTD patients^14^.”

5) Further validation of the luciferase assay to address concerns raised by Reviewers.

In Figure 1 —figure supplement 1, we have now addressed this concern. We transfected HEK293 and NSC34 cells with the antisense 35xCCCCGG plasmid that has nLuc in-frame with poly-PG. Upon Western Blotting, we detected both nLuc signal and poly-PG, suggesting nLuc is an accurate readout.

6) An evaluation of cited studies to confirm accuracy of reported information as per concerns raised by Reviewer #1.

As requested, we have added citations and also confirmed accuracy of reported information.

7) An evaluation of DPR aggregation to demonstrate relevance in disease as outlined by Reviewer #1.

To address this important point, we conducted three experiments. With immunofluorescence staining against poly-PG and poly-PR, we observed that both DPRs are present in the nucleus and cytoplasm of HEK293 cells 48h post-transfection with the poly-PG::nLuc or poly-PR::nLuc constructs (Figure 1F-G, 2H). Moreover, we quantified DPR levels in soluble and insoluble fractions (Figure 1 —figure supplement 2). The data suggest that in our transfected cells (48h after transfection) we predominantly detect DPRs (poly-PG, poly-PR) in the soluble fraction. Lastly, we observed reduced cell survival at 48 hours post-transfection of poly-PG::nLuc or poly-PR::nLuc constructs (Figure 1H-I), suggesting our constructs produce toxic DPRs from the antisense CCCCGG transcript.

Reviewer #1 (Recommendations for the authors):1. All western immunoblot data need to be quantified and statistically analyzed. In particular, quantification of the multiple molecular weight bands in Figures 2B, 3B and 4B is needed to support the authors conclusions.

We completely agree; these quantifications and statistical comparisons are now provided in Figure 3 —figure supplement 1, Figure 4 —figure supplement 1, and Figure 5 —figure supplement 1.

2. There are several issues with the data in figure 5.a. the successful knockdown of eIF2D needs to be confirmed on the protein level.

We observed efficient eIF2D knockdown at the protein level with Western blotting upon transfection with one siRNA against eIF2D (Figure 7 —figure supplement 1). This experiment was conducted in HEK293 cells. The exact same siRNA (EIF2D siRNA-1) was transfected in the iPSC-derived motor neurons from *C9ORF72* patients (Figure 7). In iPSC-derived motor neurons, we validated efficient *EIF2D* knockdown only at the mRNA level. Unfortunately, we were not able to conduct the experiment and measure eIF2D protein levels upon siRNA transfection because we used the entire amount of protein lysates for DPR detection by immunoassays.

b. multiple siRNA targeting eIF2D should be tested to confirm that effects are specific to the targeted gene versus an off-target effect.

We have now tested a second siRNA against eIF2D and results are included in Figure 7.

With the first siRNA (EIF2D-siRNA-1), we observe robust reduction (~90%) at the mRNA level of *EIF2D*, but no effects on eIF2A (Figure 7A). This experiment was conducted on one patient line (26#6).

We followed the exact same procedure (repeated transfection) for a second siRNA (EIF2D-siRNA-2) on two additional patient lines (27#11 and 40#3). Unfortunately, in one line (27#11) we did not observe EIF2D knockdown, whereas in the other line (40#3) we observed a 50% reduction in EIF2D mRNA levels (Figure 7C-D). This variability in the efficiency of siRNAs to effectively knockdown EIF2D prevented us from drawing any strong conclusions on the role of EIF2D in DPR synthesis in the context of iPSC-derived neurons. Of note, variability in siRNA efficiency is commonly observed in human cultured neurons (PMID: 31587919). Nevertheless, our siRNA experiments (two siRNAs on two different iPSC lines) took several months to complete and we feel it is important to share our observations with the community. Hence, we have decided to include them in Figure 7.

Please, see *Results section* (lines 266 – 319) entitled “Knockdown of EIF2D in human iPSC-derived motor neurons” for a detailed description of our EIF2D results in the context of IPSC-derived neurons.

c. the authors need to be clearer about their interpretation of the data on GP/PG at this point in their study. Specifically, the GP that is measured here will include both sense-GP and antisense-PG. While Zu et al. demonstrated that the antisense strand is the primary source of GP/PG (which the authors should mention in the intro or results for the reader) the sense strand will still contribute to DPR production and could mask effects. In Figure 6D & F, the authors present more compelling data to support that eIF2D is dispensable for PG production. The authors may want to consider presenting those data sooner.

We completely agree. As suggested by the reviewer, we have extensively modified the *Results section* (lines 266 – 319) entitled “Knockdown of EIF2D in human iPSC-derived motor neurons”. We now note that: (i) our immunoassay cannot distinguish between poly-PG and poly-GP, (ii) the effects could be masked by poly-GP, and (iii) have also toned down our conclusions.

As the reviewer suggested, we changed the order and now present the HEK293 cell data first (Figure 6), followed by the iPSC data (Figure 7). This is because in our HEK293 transfections we use plasmid where nLuc is in the poly-PG frame. Hence, we have an accurate readout of poly-PG synthesis from the antisense transcript (Figure 6).

3. The authors design novel C4G2:NanoLuc constructs to study antisense DPR synthesis. Further validation of the constructs and the system would improve the current study.a. Can the authors present data showing that the PR and PG bands in Figure 1D-1E co-stain with an anti-luciferase antibody since the DPR are tagged with nLuc (Figure 1A)? This would demonstrate that the luciferase signal is an effective and accurate readout for DPR production.

This is an excellent suggestion. In new Figure 1—figure supplement 1, we have addressed this concern. We transfected HEK293 and NSC34 cells with the antisense 35xCCCCGG plasmid that has nLuc in-frame with poly-PG. Upon Western Blotting, we detected both nLuc signal and poly-PG, suggesting nLuc is an accurate readout. We present these new data in the first section of Results.

We repeated the same experiment with nLuc in-frame with poly-PR. With Western blotting, we detected poly-PR but nLuc signal was extremely faint, at the limit of detection. This is likely due to three reasons:

As shown in Figure 1B-C, a luciferase assay shows that poly-PR::nLuc is expressed at much higher levels (3-fold) than poly-PG::nLuc.

b. The authors should consider insoluble versus soluble DPR when measuring DPR production. They only look at soluble DPR but the DPR levels within the insoluble fraction may differ (Quaegebeur et al. 2020; Licata et al. 2021) and this may impact the interpretation of the later data as the DPR may be enriched in the insoluble fraction and seem depleted in the soluble fraction measured using western immunoblots and the luciferase assay. Thus, can the authors demonstrate that the levels of DPR in the insoluble fraction are consistent with the soluble fraction in their system making the soluble fraction a reasonable read-out?

Following this suggestion, we repeated our transfections and 48h later isolated both soluble and insoluble fractions. We ran Western Blots for poly-PR and poly-PG and compared side-by-side DPR levels in soluble and insoluble fractions. To assess the quality of our lysates, we used H3K4me2, a reliable marker for the insoluble fraction (PMID: 25852189).

poly-PG: We did not detect any poly-PG in the insoluble fraction in NSC34 or HEK293 cells transfected with the poly-PG::nLuc construct (Figure 1 —figure supplement 2). Poly-PG was robustly detected in the soluble fraction under these experimental conditions.

poly-PR: We also did not detect any poly-PR in the insoluble fraction in HEK293 cells transfected with the poly-PR::nLuc construct (Figure 1 —figure supplement 2). We detected a negligible amount of poly-PR in the insoluble fraction in NSC34 cells transfected with the poly-PR::nLuc construct (Figure 1 —figure supplement 2). Hence, poly-PR is robustly detected in the soluble fraction – under these experimental conditions.

We present these new data in first section of Results (lines 118 – 138).

c. It is surprising that the authors do not see PA production in Figure 1 using their luciferase assay. Can the authors confirm these results using an antibody against PA? The authors should discuss why they do not see any PA production when PA is seen in patient samples.

Indeed, we do not detect poly-PA::nLuc with a luciferase assay under our specific experimental conditions, that is, 48h upon transfection of poly-PA::nLuc construct in HEK293 or NSC34 cells (Figure 1B-C). We repeated the poly-PA::nLuc transfections and ran a Western Blot with an antibody against poly-PA. Again, poly-PA is not detected (Figure 1 —figure supplement 3).

To evaluate the specificity of the poly-PA antibody, we generated a poly-PA::nLuc plasmid that has an artificial AUG start codon located at -67bp upstream from the CCCCGG repeats. This AUG is in poly-PA frame. With this construct, we detect poly-PA signal (Figure 1 —figure supplement 3).

Altogether, this result is consistent with a previous study by *Boivin et al. (EMBO J, 2020*, PMID: 31930538): these authors also did not detect poly-PA upon transfection of HEK293 cells with a different construct carrying 100 copies of the CCCCGG repeats. Similar to our study, they only detected poly-PA upon introduction of an artificial AUG initiation codon in the poly-PA reading frame. Possible reasons for not detecting poly-PA in our study and the Boivin et al. study:

(i)the transfected constructs lack intronic regions that contain the yet-to-be identified initiation codon for poly-PA translation.(ii)the cellular context (HEK293 or NSC34 cells) lacks the specific regulatory machinery (e.g., specific translation initiation factors) needed for poly-PA synthesis.

We have modified the Results (lines 110-114) accordingly to include these reasons: “Consistent with a previous study^19^, we did not detect poly-PA with luciferase assays (Figure 1B-C) and Western blotting (Figure 1—figure supplement 3) upon poly-PA::nLuc transfection. We surmise that the initiation codon for poly-PA may lie outside the 1,000 bp intronic sequence used in our construct, or that the specific regulatory machinery needed for poly-PA synthesis is lacking in the cellular context examined here (HEK293 and NSC34 cells).”

4. The authors should consider changes in cellular toxicity in their experiments as they are discussing potential impacts in human disease. Can they demonstrate that their manipulations alter cell survival or DPR-associated toxicity? what about changes in DPR aggregation (since GP/PG may not be toxic but these do aggregate and this is pathologically relevant to patients)?

Thanks for this suggestion. We found that poly-PG::nLuc or poly-PR::nLuc constructs reduced cell survival 48h after transfection (new Figure 1H-I). Notably, we observed partial rescue of the cell survival phenotype when poly-PR synthesis was specifically blocked by mutating the AUG initiation codon at -273 position to UAG (new Figure 2I). Importantly, we observed no rescue of the cell survival phenotype when poly-PG synthesis was specifically blocked by mutating the AUG initiation codon at -113 position to UAG (new Figure 5F). As the reviewer alluded, these data suggest that poly-PR is toxic and primarily responsible for the reduced cell survival phenotype.

With immunostaining, we detected poly-PR in the cytoplasm and the nucleus (Figure 1F, 2H). Importantly, mutation of the AUG initiation codon to UAG at -273 position completely blocked poly-PR synthesis – we verified this result with Western blotting against poly-PR (Figure 2B-E), a luciferase assay nLuc (Figure 2F-G), and with fluorescence immunostaining against poly-PR (new Figure 2H).

We obtained similar results for poly-PG (Figure 5). We note that immunofluorescent staining showed that poly-PG can be more easily detected in the cytoplasm compared to poly-PR (Figure 1F-G, 2H, 5E).

We have modified the Results accordingly to include all these new data.

5. The authors need to carefully check their citations and make sure the cited studies are accurately being described. Here are some examples of issuesa. Lines 56, 202-203 and throughout: the statements "… is independent of G4C2 repeats" don't make sense. The G4C2 repeat is what codes for the DPR so DPR production cannot be independent of the G4C2 sequence.

We apologize for any inaccuracies in reporting published information. We have checked again every cited publication and have extensively modified the text.

Please rephrase these sentences to better describe previous findings that translation is mediated by flanking sequence. Also, the authors may want to be more reflective when discussing this point. Specifically, the presence of an expanded G4C2 repeat alone without flanking sequence can result in DPR production in cultured cells and in flies. So while translation may be promoted by flanking sequence in the cited manuscripts, translation may also occur through RAN translation mechanisms that solely utilize the repeat. Both modes of translation may occur simultaneously in disease while the use of non-canonical and canonical start codons may be the primary contributor of DPR production.

Thank you for this suggestion which clarifies an important message of our work. We have significantly modified the first paragraph of Discussion to address this important point.

From Discussion (lines 359-370): “… it is thought that repeat-associated non-AUG (RAN) translation of poly-GA and poly-GR occurs via non-canonical CUG and AGG initiation codons, respectively, located in the intronic sequence upstream of the GGGGCC repeats ^17, 18, 19, 20, 21, 35^. Interestingly, studies in *Drosophila* and cultured cells showed that the presence of an expanded GGGGCC repeat alone, without flanking sequences, can result in DPR production^14, 45^. Hence, our findings together with those of previous studies suggest that DPR synthesis may involve at least three different modes of translation: (a) near-cognate start codon (e.g., CUG, AGG) dependent-translation for poly-GA and poly-GR from sense GGGGCC transcripts, (b) canonical AUG-dependent translation for poly-PR and poly-PG synthesis from antisense CCCCGG transcripts, and (c) DPR synthesis may also occur through RAN translation mechanisms that solely utilize the repeat. It is conceivable that all three modes of translation may occur simultaneously in disease, but the use of non-canonical and canonical initiation codons may be the primary contributor of DPR production”.

b. Line 224: the sentence describing Goodman et al. is incorrect. This study utilized a fly model that carried a ~100bp of the 5' flanking sequence found in patients and thus was not a G4C2 repeat alone as described. Further, eIF4B/H was not show to "directly bind" the G4C2 RNA but rather there was a correlation in eIF4B/H downregulation and reduced GR-GFP production with no impact on non-repeat transcripts.

Thank you. We have removed this sentence from the Discussion. In the revised manuscript, eIF4B/H are only mentioned in the Introduction.

c. Line 42: it may be more transparent to change "thought to" to "predicted to" for readers unfamiliar with the topic. "Thought to" indicates that there is evidence to support all 3 potential modes of toxicity vs predicted to which clearly indicates these are hypothetical scenarios. However, the contribution of reduced C9orf72 protein to disease is not well established. Further, this point needs to be de-emphasized in favor of gain-of-function modes of toxicity by listing it as the last potentiality versus the first potentiality.

We made the requested changes in first paragraph of Introduction.

d. Line 68: this is not an exhaustive list of RAN translation factors identified for G4C2. There is also DHX36 (Tseng et al. 2021) and eIF4B/eIF4H (Goodman et al. 2019; supported by data in Linsalata et al. 2019). There may be others as well. Please add missing information.

We agree. Some of this information was included in Discussion. We conducted a thorough literature search and the revised Introduction now includes a comprehensive list of reported factors necessary for DPR synthesis.

Lines 75-79: “Research efforts have uncovered a number of proteins that act at different steps of DPR synthesis: RNA helicases (eIF4A, DHX36, and DDX3X)^17, 27, 28^, proteins of the eIF4F complex (eIF4A, eIF4B, eIF4E, eIF4H) ^17, 29, 30, 31^, small ribosomal protein subunit 25 (RPS25)^32^, ribosome quality control protein ZNF598^33^, and eukaryotic translation initiation factors (DAP5^21^, eIF2A^20^, eIF3F^34^, eIF2D^35^, and eIF2D co-factors DENR and MCTS^-136^).”

e. Please add citations after the following statements and check for additional instances where citations are missing.i. Line 43: citations for loss of C9orf72 potentially contributing to disease.

We added the missing references.

ii. Line 52: citations for CUG for GA and AGG for GR.

We added the missing references.

iii. Line 58: citation for PG synthesis using AUG.

Done.

iv. Line 63: citation for the failed clinical trial.

Unfortunately, the results of the two clinical trials (Biogen and WAVE) were only communicated with a press release. We do cite studies that used an ASO to target the sense GGGGCC transcript in a C9ORF72 patient (PMIDs: 34949835, 35589711, 35592494); these studies are highly relevant to the ASO clinical trials.

Lines 71-72: “Lastly, two recent ALS clinical trials that specifically targeted the production of DPRs from the sense transcript failed^24, 25, 26^”.

We also checked for additional instances where citations were missing. We found 2 instances related to RNA and DPR toxicity in Introduction, and have now added more references.

Reviewer #2 (Recommendations for the authors):In all the luciferase reporter figures, the relative values of all the constructs should be provided. It is expected that mutating AUGs will drastically reduce the translation level. However, whether the mutant reporter still have higher luciferase (by low efficient RAN translation) than no repeat control was not shown. This information needs to be included.

We now include in all luciferase quantifications the relative values (Figure 1B-C, 2F-G, 3C-D, 4C-D, 5C-D). In most cases, statistical analysis shows that the mutant reporter has luciferase levels comparable to the control construct (DC9) that bears no CCCCGG repeats. The fact that AUG mutation did not completely suppress DPR translation to the exact same level with the control (DC9) construct may have to do with rules that govern stringency of start codon selection, as described in a paper concerning the fungus *Neurospora crassa* (PMID: 23396971).

It has not been resolved what is the contribution of GP from sense and antisense strands. It has been shown in previous publications that reduction of sense repeats can significantly decrease the GP level. Therefore, there is not sufficient evidence to claim that knocking down eIF2D had no effect on GP level is due to the AUG translation of antisense repeats.

We completely agree. Our experiments in HEK293 cells were done with a poly-PG::nLuc construct, where nLuc is in the poly-PG frame (Figure 6). In HEK293 cells, KO or knockdown of eIF2D did not reduce poly-PG::nLuc expression. Please, see revised Results section (lines 188-203) entitled “eIF2D does not control poly-PR and poly-PG synthesis from the antisense transcript”.

In the experiment with iPSC-derived motor neurons (Figure 7), we toned down this claim because our DPR detection method cannot distinguish between poly-PG (from antisense RNA) and poly-GP (from sense RNA). As the reviewer suggested, we also included in that section of the Results the literature that suggests that “It has not been resolved what is the contribution of GP from sense and antisense strands”. Please, see revised *Results section* entitled “Knockdown of EIF2D in human iPSC-derived motor neurons”.

Lines 216 – 224: “We caution though that our immunoassay does not distinguish between poly-PG produced from the antisense transcript and poly-GP from the sense transcript (Figure 7B). Hence, a mild effect upon eIF2D knockdown on poly-PG (from antisense transcript) can potentially be masked by poly-GP (from sense transcript). Of note, PG/GP inclusions in brain tissue of C9ORF72 ALS/FTD patients contain ~80% of poly-PG from the antisense transcript and ~20% of poly-GP from the sense transcript^14^. However, other studies indicate that the exact contribution of sense poly-GP and antisense poly-PG C9ORF72 ALS/FTD has not been resolved^25, 26, 46^. Hence, our data hint that eIF2D may not affect poly-PG synthesis from the antisense CCCCGG transcript.”

Only one isogeneic iPSC line was used in the study. 3-5 individual lines should be used to demonstrate the conclusion.

We conducted new experiments. To test the role of eIF2D, we have now used three published iPSC lines from *C9ORF72* carriers, as well as their isogenic control lines which had CRISPR/Cas9-mediated deletion of expanded GGGGCC repeats. As we describe in Results, we only achieved (~90%) efficient EIF2D knockdown in neurons derived from one iPSC line from a *C9ORF72* carrier. In the remaining two lines, we witnessed variability in our siRNA approach to knockdown EIF2D.

This variability in the efficiency of siRNA to effectively knockdown EIF2D prevented us from drawing any strong conclusions on the role of EIF2D in DPR synthesis in the context of iPSC-derived neurons. Of note, variability in siRNA efficiency is commonly observed in human cultured neurons (PMID: 31587919). Nevertheless, our siRNA experiments (two siRNAs on two different iPSC lines) took several months to complete, and we feel it is important to share our observations with the community. Hence, we have decided to include them in Figure 7, but refrain from drawing strong conclusions on the role of EIF2D in DPR synthesis in the context of neurons derived from iPSCs of *C9ORF72* carriers.

Please, see *Results section* (lines 205 – 239) entitled “Knockdown of EIF2D in human iPSC-derived motor neurons” for a detailed description of our EIF2D results in the context of IPSC-derived neurons.

To prove the endogenous DPR production from the antisense repeats starts from AUG, the regions containing the AUG codons should be mutated/deleted to assess how it affects the DPR level.

To specifically address this issue, the lab of one of the authors (Dr. Fen-Biao Gao) is currently preparing a separate manuscript. Specifically, the Gao lab tried multiple times to mutate the AUG codons in the poly-PG frame with CRISPR/Cas9 gene editing, but the region is very rich in GC content and the approach was unsuccessful. In this upcoming separate manuscript by the Gao lab, the authors were successful in generating a deletion of the intronic region that contains multiple AUG codons.

In Figure 5, eIF2D protein level should be measured by western blotting.

We observed efficient eIF2D knockdown at the protein level with Western blotting upon transfection with siRNA against eIF2D (Figure 6 —figure supplement 1). This experiment was conducted in HEK293 cells. The exact same siRNA was transfected in the iPSC-derived motor neurons from C9ORF72 patients (Figure 7). In iPSC-derived motor neurons, we validated efficient eIF2D knockdown at the mRNA level. Unfortunately, we were not able to conduct the experiment and measure eIF2D protein levels upon siRNA transfection because we used the entire amount of the protein lysates for DPR detection.

In figure 6, the GA level should be measured as the positive control.

We have now included this positive control (Figure 6). We observe reduced luciferase activity in EIF2D KO or shRNA conditions, albeit statistical significance is only observed with the EIF2D shRNA. We surmise there is genetic compensation (transcriptional adaptation) when EIF2D gene activity is completely removed in HEK293 cells, a phenomenon that recently received attention (PMID: 32816903).

[Editors’ note: what follows is the authors’ response to the second round of review.]

The manuscript has been improved but there are some remaining issues that need to be addressed, as outlined below:Edits are needed to address concerns raised by Reviewer #3. In particular, the new data on eIF2D as a GA-specific RAN translation factor has lead to some conflicting results with previous studies. This conclusion needs to be more fairly presented in the text, especially in the abstract and discussion.Reviewer #3 (Recommendations for the authors):The authors included new data to address most of the concerns and significantly improved the manuscript. However, as the reviewer was concerned, some of the conclusions cannot be supported when tested in more patient lines.

Glad to see that reviewer #3 finds the revised manuscript improved.

In the two additional patient iPSN lines, eIF2D knockdown did not reduce the GA level, but instead increased the GA level. Although the authors reasoned that the siRNA did not work well in the other two lines, the line 40#3 has good knockdown, and the upregulation of GA in two out of three lines is hard to interpret.

We agree with the reviewer. We therefore mention in Results (lines 273-276) that we cannot draw any general conclusions on the role of EIF2D in DPR synthesis in the context of motor neurons derived from different iPSC lines of *C9ORF72* carriers.

Furthermore, the data in Figure 6 also did not show a big effect of reducing eIF2D (both KO and shRNA) on poly-GA level. Overall, the data cannot support the claim that eIF2D is generally required for poly-GA translation from a CUG initiation codon.

We now make clear in the text that the reason for this is likely technical. In Sonobe et al., 2021 (PMID: 34654821), we observed a strong effect on poly-GA level in NSC34 cells upon EIF2D shRNA. For that experiment, we introduced the 75 GGGGCC repeats in the context of a monocistronic construct. In that same study, we also conducted a similar experiment in HEK293 cells using a bicistronic construct and observed a strong effect on poly-GA. In the current manuscript, we used a monocistronic construct in HEK293 cells and observed a small effect on poly-GA.

To communicate more precisely our findings on EIF2D, we edited the Abstract, Results and Discussion.

For the antisense repeat translation, the data in the manuscript showed the AUG can be used if the transcription starts more upstream. But without data from endogenous mutagenesis, this cannot be validated. The writing needs to be modified to clarify this point.

We have modified the Discussion (lines 314-320) to emphasize that future endogenous mutagenesis studies are needed to solidify our in vitro findings that rely on plasmids:

“The AUG initiation codons we identified as necessary for either poly-PR or poly-PG synthesis are predicted to be included in the endogenous antisense CCCCGG transcript based on 5’ Rapid Amplification of cDNA Ends (RACE) analysis on brain samples of C9ORF72 ALS/FTD patients^14^. Nevertheless, endogenous mutagenesis of these codons – in the native genomic context of the *C9ORF72* locus – is needed in the future to further test the validity of our findings.*”*

There is no Figure 7—figure supplement 1.

We regret the omission of this file, which we have now provided. The reviewer had asked us to evaluate knockdown of eIF2D at the protein level.

In Figure 7 —figure supplement 1, we show efficient EIF2D knockdown at the protein level with Western blotting upon transfection with one siRNA against EIF2D, referred in the text as “EIF2D siRNA-1”. This experiment was conducted in HEK293 cells. The exact same siRNA (EIF2D siRNA-1) was transfected in iPSC-derived motor neurons from *C9ORF72* patients (Figure 7). In iPSC-derived motor neurons, we validated efficient *EIF2D* knockdown only at the mRNA level. Unfortunately, we were not able to conduct the experiment and measure EIF2D protein levels upon siRNA transfection because we used the entire amount of protein lysates for immunoassays to detect DPR levels.